# STORYCODER: BRIDGING NARRATIVES AND FORMALITY FOR CODE GENERATION IN LLMS

## ABSTRACT

The code generation capabilities of large language models stem largely from pre-training on structured code patterns and their strong context-based reasoning abilities. However, previous research on code generation has primarily focused on using short, fragmented, instruction-like prompts, which often fail to encourage contextual understanding. Inspired by the way humans organize fragmented information into coherent explanations, we propose a new method that reformulates coding problems as natural language narratives to promote integrative thinking. To this end, we introduce STORYCODER, a framework that reformulates code generation prompts into narrative text. Our results show that rich contextual expressions in natural language can enhance code generation performance and lead models to adopt consistent and structured problem-solving strategies. We quantitatively demonstrate that our method provides integrative information not captured by simple rephrasings and guides models to adopt correct algorithms and implementation strategies, thereby improving code generation performance. Experiments on three benchmarks, HumanEval, CodeForces, and LiveCodeBench, show an average improvement of $28.6\%$ in the precision of zero-shot pass@10.

## 1 INTRODUCTION

Problem-solving ultimately depends on how clearly the information is conveyed and understood. Effective problem solving may require both the solver's capability to interpret information and the way the problem is framed (Vessey, 1991; Kelton et al., 2010). However, in practice, task descriptions are incomplete or ambiguous, forcing solvers to infer missing details from context. These gaps are especially challenging in complex tasks that demand contextual understanding or multi-step reasoning. Large language models (LLMs) face the same difficulty: their performance depends not only on internal reasoning but also on how effectively the task is specified and interpreted (Laban et al., 2025). In this work, we investigate how to improve the delivery and interpretation of information in LLMs, focusing on code generation tasks. Programming tasks are particularly suitable for this study: they are built on logically distinct structures, and their solutions can be explicitly validated using test cases (Wang et al., 2025; Light et al., 2025).

We introduce STORYCODER, a narrative-based prompting method that transforms short, instruction-like problem statements into coherent natural language. This design is grounded in cognitive science findings that humans comprehend and reason more effectively by organizing fragmented conditions into coherent mental models and using analogical structures to facilitate deeper reasoning (Johnson-Laird, 1983; Gentner, 1983; Holyoak & Lu, 2021). In this framework, models identify the appropriate algorithm that will form the logic of the code, align it with a suitable narrative genre, and reformulate it into a story with three sections: task overview, constraints, and example input/output (Figure 1.) By connecting fragmented conditions into coherent descriptions, narratives help LLMs capture context and follow more structured reasoning. We evaluate STORYCODER across closed-source and open-source models on three benchmarks—HumanEval, CodeForces, and LiveCodeBench. Extensive experiments demonstrate consistent performance gains, with an average improvement of $28.3\%$p in zero-shot pass@10 accuracy.

Beyond overall accuracy, we find that narrative prompts substantially increase the likelihood of selecting the correct algorithms and reduce implementation errors. In contrast, when narratives are expressed in mismatched genres, performance drops significantly. This suggests that LLMs

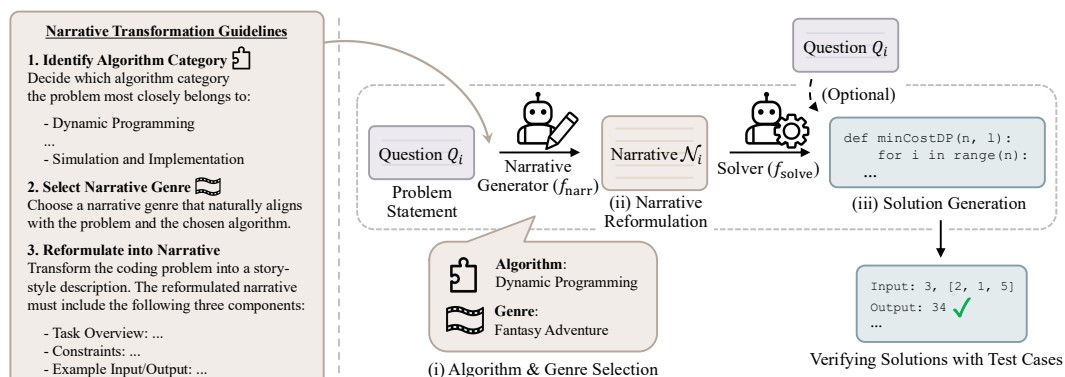

Figure 1: **Overview of STORYCODER framework.** Given a question $Q_i$, (i) model first identifies an algorithmic category and selects a narrative genre, then (ii) reformulates the problem into a structured narrative $\mathcal{N}_i$ consisting of task overview, constraints, and example input/output, and (iii) passes the narrative (optionally with $Q_i$) to a solver model to generate code solutions, which are then verified with test cases.

have an inherent ability to understand narrative structures that align well with tasks such as code generation. These observations support our hypothesis that narratives are a natural and effective tool to encourage integrative reasoning.

Our key contributions are as follows:

- We propose STORYCODER, a narrative-based prompting method for code generation that reformulates fragmented prompts into coherent descriptions.

- We demonstrate consistent empirical improvements across diverse models and benchmarks, achieving a 28.6%p average gain in zero-shot pass@10 accuracy.

- We provide quantitative analyses showing that narrative reformulation guides LLMs in choosing the correct algorithms and reducing implementation errors.

## 2 RELATED WORK

**Structured prompt engineering for code generation.** Recent work in code generation has increasingly focused on structuring prompts to improve the reasoning process of LLMs. Some approaches introduce explicit intermediate steps, such as control structures or modular subcomponents, to guide program synthesis more reliably (Li et al., 2025a; Le et al., 2024; Huang et al., 2023). Recent work demonstrates that language-based input–output patterns can enable structured and verifiable reasoning in code generation (Li et al., 2025b). PECC (Haller et al., 2024) further explores how narrative-embedded problem descriptions influence a model's ability to extract requirements from prose-style coding tasks, highlighting the role of structured natural language in shaping code understanding. These strategies typically aim to align the internal reasoning process of the model with syntactic correctness and functional accuracy. In addition, continued pretraining on mathematical code helps models better handle symbolic expressions and abstract logic in programming tasks (Lu et al., 2025). While these approaches rely on explicitly structured reasoning or fine-tuned feedback signals, our method STORYCODER adopts a narrative-based prompt design that encourages integrative understanding without requiring explicit modular decomposition.

**Prompt reformulation and test-time reasoning in LLMs.** Another line of research examines how rephrasing prompts at test time affects the way LLMs reason on a task. Chain-of-thought prompting, especially when combined with self-consistency, helps models explore diverse reasoning paths and improves robustness through output aggregation (Wei et al., 2022; Wang et al., 2023). Beyond explicit reasoning steps, rephrasing task instructions influences how the model interprets the problem (Fu et al., 2024; Zhou et al., 2024). Story-of-Thought (SoT) (Sadiri Javadi et al., 2025) extends this perspective by converting problems into narrative-style descriptions for multiple-choice commonsense reasoning, showing that narrative framing can modulate task interpretation. Building on this insight, our method reformulates prompts into coherent task-grounded narratives that embed conditions, intent, and examples within a unified structure tailored for code generation.

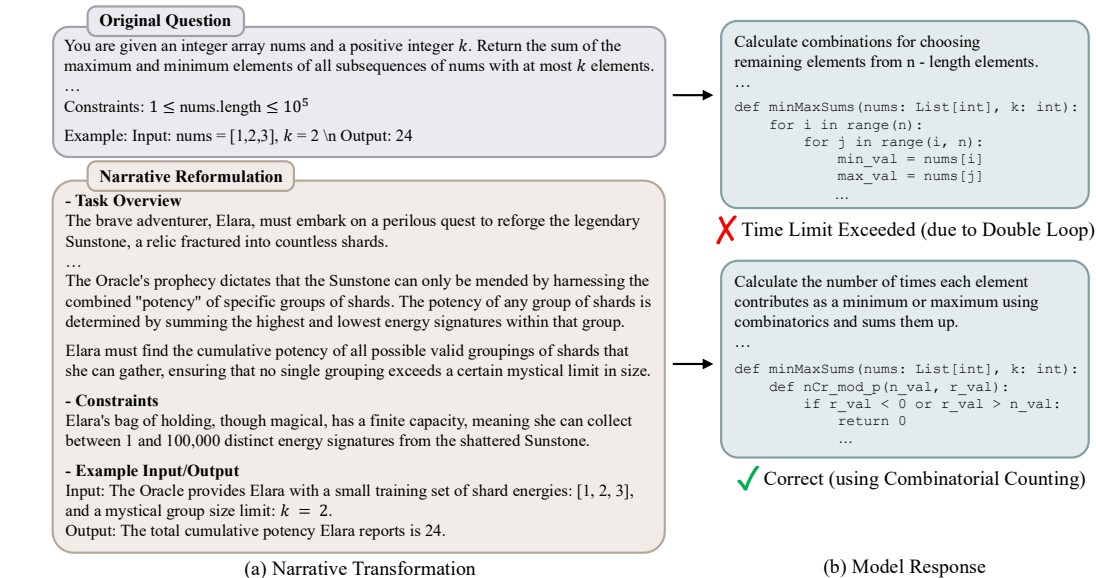

(a) Narrative Transformation

(b) Model Response

Figure 2: **Example of narrative reformulation.** The narrative representation bridges problem description and model reasoning, guiding the model from inefficient non-optimal solutions toward algorithmic strategies.

## 3 STORYCODER: REFORMULATING QUESTION INTO NARRATIVES

### 3.1 CONVENTIONAL BASELINES FOR CODE GENERATION

Early evaluations of code generation often used similarity-based metrics (e.g., CodeBLEU), but these failed to capture functional correctness. Recent benchmarks adopt execution-based evaluation, leading to the pass@$k$ metric (Chen et al., 2021). Within this setting, following baselines are considered: Repeated Sampling, Paraphrasing and Chain-of-Thought (CoT) prompting.

**Repeated Sampling** is the simplest way to improve pass@$k$ performance. The model is asked to generate multiple outputs for the same input, typically by stochastic sampling with high temperature. With more candidates in the pool, the chance of including at least one correct solution increases.

**Paraphrasing** is another simple baseline that reformulates the original problem statement by altering surface expressions without affecting its meaning (Zhou et al., 2024). This provides a variant of the prompts while leaving the fundamental representation of the task unchanged.

**Chain-of-Thought (CoT)** is another approach that encourages the model to structure its problem-solving process (Wei et al., 2022; Huang et al., 2023). Instead of directly producing code, the model is prompted to generate intermediate reasoning steps before the final output. This technique introduces explicit reasoning into the generation process, helping the model reflect on its problem-solving steps.

### 3.2 NARRATIVE REFORMULATION FOR DEEPER COMPREHENSION

The approaches discussed in Section 3.1 are useful, but they are limited in expanding the solution space, involve surface-level changes, or fail to reflect actual model reasoning (Turpin et al., 2023). To overcome these limitations, we propose a narrative reformulation framework that enables the model to deeply understand input representations. This framework helps the model understand the task through structured analogies and metaphors, and it is grounded in findings from cognitive science.

We construct a framework that reformulates code generation questions into a narrative format in three stages as shown in Figure 1: (i) for the $i$-th question $Q_i$, choosing an appropriate algorithmic category $a_i$ and a narrative genre $g_i$; (ii) rewriting $Q_i$ as a structured narrative $\mathcal{N}_i$ with three parts: task overview, constraints, and example input/output; and (iii) solving the task using $\mathcal{N}_i$. Here,

the algorithm $a_i$ denotes the algorithmic category that the model judges to be the most appropriate for the given problem, chosen from the eight predefined categories in Figure B.1. The genre $g_i$ denotes the narrative style, selected freely by the model to align with the problem and the chosen algorithm $a_i$. We generate $N$ narrative variants for each question $Q_i$ and denote the index of each reformulation variant by $j \in \{1, \ldots, N\}$. Note that generating these narrative variants differ from simply drawing multiple solutions, as each narrative provides a distinct perspective and plot that broadens the model's representational space for interpreting and reasoning about the task. Formally, the overall pipeline can be described as follows:

$$\{a_i^j,\ g_i^j,\ \mathcal{N}_i^j\}_{j=1}^N = f_{\text{narr}}(Q_i), \qquad \text{Ans}(\mathcal{N}_i^j) = f_{\text{solve}}(\mathcal{N}_i^j), \tag{1}$$

where $f_{\text{narr}}$ is the generator model that formulates narratives and $f_{\text{solve}}$ is the solver model that generates $\text{Ans}(\cdot)$. We refer to the case where $f_{\text{narr}}$ and $f_{\text{solve}}$ are the same model as the self-solving setting, and to the case where they differ as the cross-model setting. Note that $\mathcal{N}_i^j \sim P(\cdot \mid a_i^j, g_i^j)$, since the narrative $\mathcal{N}_i^j$ is conditioned by $a_i^j$ and $g_i^j$ chosen by $f_{\text{narr}}$.

In stage (ii), unlike the prior work (Sadiri Javadi et al., 2025), we carefully design the narrative components for programming tasks, where precise format and formal constraints are required. To ensure that reformulated problem preserve both narrative coherence and computational strictness, we divide a narrative $\mathcal{N}_i^j$ into three parts:

- **Task Overview** $(\text{TO}_i^j)$: presents the coding objective within a narrative frame, integrating scattered conditions into a coherent system that guides comprehension and reasoning.

- **Constraints** $(\text{C}_i^j)$: reframes input ranges, time limits, and operational rules as natural restrictions in the story, allowing the model to internalize constraints within the narrative space.

- **Example Input/Output** $(\text{E}_i^j)$: integrates sample test cases into contextual scenarios, aligning input/output examples with the story structure, so that bridging between concrete problem solving region and the narrative space.

This three-part structure is grounded in well-established findings from cognitive science. First, situating the problem in a **Task Overview** aligns with the theory of mental models, which emphasizes that humans comprehend and reason by constructing coherent representations of situations (Johnson-Laird, 1983). Second, articulating **Constraints** within the narrative makes the rules clear, so that readers do not have to think hard to integrate fragmented information. This reduces unnecessary mental effort and helps readers focus on solving the problem (Chandler & Sweller, 1991). Finally, integrating **Example Input/Output** into narrative situations takes advantages of the role of analogy in facilitating understanding and transfer, mapping abstract problem structures onto given scenarios (Gentner, 1983; Holyoak & Lu, 2021).

The transformation example is shown in Figure 2 and Figure B.2. Together, these three components of the narrative $\mathcal{N}_i^j = \{\text{TO}_i^j,\ \text{C}_i^j,\ \text{E}_i^j\}$ standardize the reformulation process, ensuring that all essential details of the original question of code generation are preserved while also allowing cognitive principles to be naturally integrated into the narrative structure. Please refer to Appendix B.3 for the full guidelines and transformation examples.

## 4 EXPERIMENTS

### 4.1 EXPERIMENTAL SETTINGS

**Models.** We evaluate a total of 11 models of varying sizes. Among open-source models, we use the instruction-tuned versions of Deepseek-Coder 6.7B (Guo et al., 2024), Deepseek-Coder-V2-Lite (Zhu et al., 2024), Llama-3.1 8B (Grattafiori et al., 2024), Gemma-2 9B and 27B (Team et al., 2024), Qwen-2.5-Coder 7B and 32B (Hui et al., 2024), and Mistral-Small 24B (Mistral AI, 2025). For readability, we omit 'Instruct' from all the model names below. For closed-source models, we include Claude-3.5-Haiku (Anthropic, 2024), Gemini-2.5-Flash (Comanici et al., 2025), and GPT-4.1-mini (OpenAI, 2025). All code generation is performed with a temperature setting of 0.2.

**Dataset.** We evaluate on three benchmarks: HumanEval (Chen et al., 2021), a hand-crafted dataset with function signatures and docstrings; LiveCodeBench (Jain et al., 2024), a large-scale dataset

Table 1: **Pass@10 performance** on three benchmarks. The upper part of the table reports closed-source models, while the lower part reports open-source models. Narrative prompting consistently improves performance over repeated sampling (RS) across all benchmarks and models. Pass@$k$ performance curves for all models and benchmarks are provided in Appendix A.1.

| Model | HumanEval | | LiveCodeBench | | CodeForces | |
|---|---|---|---|---|---|---|
| | RS | Narrative | RS | Narrative | RS | Narrative |
| Gemini-2.5-Flash | **96.19** | **96.19** | 49.71 | **57.14** | 45.48 | **65.97** |
| GPT-4.1-mini | **94.29** | **94.29** | 47.43 | **56.57** | 31.23 | **49.61** |
| Claude-3.5-Haiku | 85.71 | **94.29** | 33.71 | **38.29** | 46.45 | **50.51** |
| Average | 92.06 | **94.92** | 43.62 | **50.67** | 41.05 | **55.36** |
| DSCoder 6.7B | 82.86 | **90.48** | 22.29 | **27.43** | 13.12 | **18.01** |
| DSCoder-V2-Lite | 78.10 | **93.33** | 28.57 | **34.29** | 25.16 | **33.14** |
| Llama-3.1 8B | 79.05 | **81.90** | 21.14 | **27.43** | 8.11 | **19.08** |
| Gemma-2 9B | 63.81 | **82.86** | 20.00 | **26.29** | 11.69 | **21.39** |
| Gemma-2 27B | 76.19 | **87.62** | 27.43 | **34.29** | 22.38 | **30.97** |
| Qwen-2.5-Coder 7B | 89.52 | **93.33** | 26.86 | **33.14** | 19.31 | **26.74** |
| Qwen-2.5-Coder 32B | 92.38 | **94.29** | 30.86 | **40.00** | 15.08 | **27.10** |
| Mistral-Small 24B | 88.57 | **94.29** | 33.71 | **34.86** | 29.23 | **42.87** |
| Average | 81.31 | **89.76** | 26.36 | **32.22** | 18.01 | **27.41** |

covering various programming problems from multiple platforms; and CodeForces (Mirzayanov, 2010), a collection of real-world algorithmic problems from competitive programming. For HumanEval, we exclude questions that are invalid or containing incorrect sample input/output, resulting in a filtered set of 105 questions. For LiveCodeBench, we use the 175 questions from release-v6 and confirm that all are sourced from AtCoder (Ueda & Inc., 2012) or LeetCode (Tang, 2015). For CodeForces, we apply filtering based on question length and difficulty, since evaluating over 10,000 problems is impractical, yielding a final set of 265 questions. For detailed filtering criteria and additional results, see the Appendix B.2.

**Metric and Evaluation.** We report the results using the pass@$k$ metric, which measures the probability that at least one out of $k$ generated solutions is correct. Following HumanEval, we consider a generated solution to be correct only when it passes all test cases (see Appendix B.1 for details).

In our experiments, we set $N = 5$ narrative variants per question. For the Narrative setting in the main paper, we aggregate ten total responses per question: five narrative-only variants $\{\mathcal{N}_i^j\}_{j=1}^5$ and five narrative concatenated with the original question $\{\mathcal{N}_i^j, Q_i\}_{j=1}^5$. The narrative-only form and its concatenation with the original question are two equivalent realizations of the same reformulation pipeline. The original question serves as a supplementary context within the narrative, and aggregating the two forms therefore reflects performance over this unified input space. To ensure fairness, the repeated sampling baseline generates 10 samples per question, matching the total number of samples used in the Narrative setting. STORYCODER also shows consistent improvements on each individual set of five variants, as reported in Appendix A.2.

## 4.2 EXPERIMENTAL RESULTS

Table 1 presents pass@10 results on three coding benchmarks. For closed-source models, we adopt a self-solving setting where $f_{\text{narr}} = f_{\text{solve}}$, while for open-source models, we adopt a cross-model setting, i.e., the narratives are generated by Gemini-2.5-Flash ($f_{\text{narr}}$) and solved by each open-source model ($f_{\text{solve}}$). Narrative prompting consistently outperforms the repeated sampling, demonstrating its general effectiveness for code generation. Improvements appear not only on HumanEval but also on more challenging benchmarks such as CodeForces and LiveCodeBench. Additionally, the pass@$k$ curves in Appendix A.1 show that narrative prompting outperforms the baseline consistently as $k$ increases. Additional experimental results on paraphrasing and chain-of-thought variants are provided in Appendix A.2.

Table 2: **Proportion of valid narratives** (i.e., without abnormal repetition or empty content) where each model follows the transformation guidelines properly. Note that for the DeepSeek and Qwen families, we used their base instruction-tuned versions (DeepSeek-V2-Lite-Chat (Zhu et al., 2024), Qwen2.5-7B/32B-Instruct (Yang et al., 2025)) rather than the variants further fine-tuned for coding tasks because the transformation tasks are closer to natural language tasks.

| | DeepSeek | Gemma | | Llama | Mistral | Qwen2.5 | |
|---|---|---|---|---|---|---|---|
| Model | V2-Lite | 27B | 9B | 3.1 8B | Small-24B | 32B | 7B |
| Valid (%) | 68.11 | 95.96 | 51.74 | 36.66 | 86.90 | 76.59 | 37.80 |

Table 3: **Pass@$k$ performance of open-source models.** N-Q, N-M, and N-G correspond to $f_{\text{narr}}$ using Qwen2.5 32B Instruct, Mistral-Small 24B Instruct, and Gemma 2 27B Instruct, respectively. Repeated sampling (RS) is evaluated with standard pass@10. Narrative-based scores (N-Q, N-M, N-G) are computed as pass@$k$ with $k = n \in [8, 10]$, using the maximum number of valid narrative samples available for each problem, thereby making the evaluation closer to the RS setting. Here, 'Used Samples Ratio' denotes the proportion of problems used for scoring.

| | HumanEval | | | | LiveCodeBench | | | | CodeForces | | | |
|---|---|---|---|---|---|---|---|---|---|---|---|---|
| | RS | N-Q | N-M | N-G | RS | N-Q | N-M | N-G | RS | N-Q | N-M | N-G |
| Used Samples Ratio | 100.0 | 64.42 | 97.14 | 95.24 | 100.0 | 68.00 | 79.77 | 100.0 | 100.0 | 65.49 | 86.5 | 98.9 |
| DSCoder 6.7B | 82.86 | **92.54** | 86.27 | 87.0 | 22.29 | 19.33 | **23.19** | 22.86 | 13.12 | 12.34 | **13.14** | 12.92 |
| DSCoder V2 Lite | 78.10 | 91.04 | **92.16** | 90.0 | 28.57 | 28.57 | **31.88** | 29.71 | 25.16 | 30.00 | **32.25** | 28.17 |
| Llama 3.1 8B | 79.05 | **85.07** | 81.37 | 84.0 | 21.14 | 22.69 | **25.36** | 24.0 | 8.11 | 14.34 | **14.56** | 14.20 |
| Gemma 2 9B | 63.81 | **86.57** | 85.29 | 85.0 | 20.00 | 19.33 | 22.46 | **22.86** | 11.69 | 17.23 | 17.36 | **18.36** |
| Gemma 2 27B | 76.19 | 88.06 | **88.24** | 84.0 | 27.43 | 25.21 | 26.09 | **28.57** | 22.38 | 27.86 | **30.18** | 27.06 |
| Qwen 2.5 Coder 7B | 89.52 | 94.03 | **94.12** | 90.0 | 26.86 | 28.57 | **33.33** | 29.71 | 19.30 | 21.02 | **21.82** | 21.78 |
| Qwen 2.5 Coder 32B | 92.38 | **95.52** | 94.12 | 93.0 | 30.86 | 32.77 | 36.96 | **40.57** | 15.08 | 23.40 | 24.02 | **26.06** |
| Mistral-Small 24B | 88.57 | **92.54** | 92.16 | 89.0 | 33.71 | 31.93 | **35.51** | 32.57 | 29.23 | 32.23 | **34.68** | 33.52 |
| Average | 81.31 | **90.67** | 89.22 | 87.75 | 26.36 | 26.05 | **29.35** | 28.86 | 18.01 | 22.30 | **23.50** | 22.76 |

To evaluate the self-solving setting for open-source models, we initially considered evaluating them in a strict self-solving setting (i.e., $f_{\text{narr}} = f_{\text{solve}}$). However, as shown in Table 3, open-source models often fail to reliably follow the required narrative format, resulting in substantial differences in valid narrative ratios. To enable a more equitable comparison, we select the top-3 open-source models with the highest valid narrative rates (Qwen 2.5 32B Instruct, Mistral-Small 24B Instruct, and Gemma 2 27B Instruct) and use them as narrative generators ($f_{\text{narr}}$). Although this setup is not strict self-solving, this relaxed configuration allows us to assess whether narrative reformulation improves performance when narratives are produced by open-source models rather than relying on a strong generator. For detailed filtering criteria and examples of invalid narratives, refer to Appendix B.4.

The results of pass@$k$ performance are shown in Table 3. Even in the self-solving setting of open-source models, STORYCODER consistently improves performance across all benchmarks. On HumanEval, narratives generated by Qwen 2.5 32B (N-Q) consistently yield the strongest improvements across solvers. In contrast, on both LiveCodeBench and CodeForces, Mistral-Small 24B narratives (N-M) achieve the highest performance, showing more stable gains than Qwen- or Gemma-based narratives. These results suggest that the effectiveness of narrative reformulation depends on benchmark difficulty: Qwen 2.5 32B narratives are particularly effective for relatively easier tasks such as HumanEval, whereas Mistral-based narratives provide more reliable improvements on more challenging benchmarks such as LiveCodeBench and CodeForces.

## 5 DISCUSSION

In experiments with code generation benchmarks, systematic performance analysis requires extracting and examining algorithmic sketches, which are critical components for achieving successful code generation. To this end, we design an experimental setup to extract and evaluate algorithms

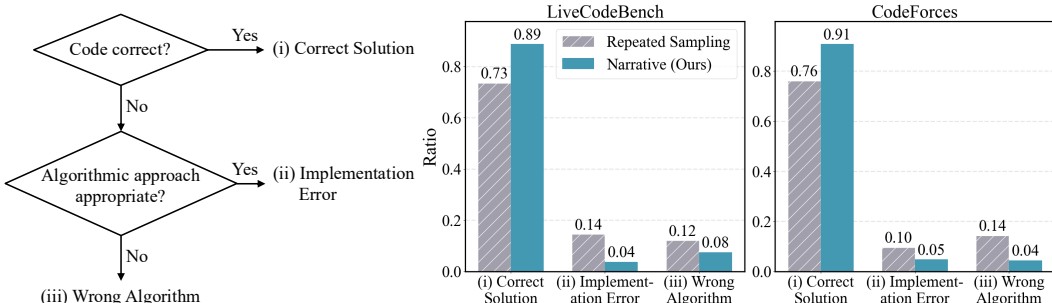

Figure 3: **Decomposition of model outputs** into (i) correct solution, (ii) implementation errors, and (iii) incorrect algorithm choice under narrative vs. original prompts. Narrative prompts increase correct solution while reducing both implementation errors and misaligned choices.

from model outputs. Given a code solution generated by the solver model $f_{\text{solve}}$, we query another model instance $f_{\text{alg}}$ to identify the algorithm underlying the solution. This procedure is similar to back-translation in machine translation, where target–source language pairs are generated for self-training (Sennrich et al., 2016; Wang et al., 2025). We apply this process to both the original and narrative versions of each problem, enabling two key comparisons: between the back-translated algorithms from model outputs and the algorithms incorporated into the narratives $a_i^j$ defined in the main experiments (Section 5.1;) and between the back-translated algorithms of wrong solutions and the golden algorithms $a_i^*$ derived from correct solutions (Section 5.2.) If not otherwise specified, experiments are conducted with Gemini-2.5-Flash.

## 5.1 NARRATIVES IMPROVE PROBLEM COVERAGE AND ALGORITHM AGREEMENT

We first compare the back-translated algorithms extracted from all model outputs with the algorithms incorporated into the narratives $a_i^j$ produced in Section 4. We define coverage as the proportion of problems for which at least one correct solution is generated, which is equivalent to pass@$k$ when $k = n$, i.e., when $k$ matches the number of generated samples per problem. We also define agreement ratio as the fraction of correct solutions whose underlying algorithms match the algorithm that the model initially predicted when observing the original problem.

Figure 4 shows that narrative reformulation improves both properties. Along the x-axis, pass@10 (coverage) increases on challenging benchmarks such as LiveCodeBench and CodeForces. This indicates that narratives expand the feasible solution space, especially for harder tasks. Along the y-axis, the agreement ratio increases consistently across all benchmarks, demonstrating that the algorithm selected initially is faithfully reflected in the generated solutions. Together, these results show that *narratives enable broader coverage and strengthen algorithmic consistency*, ensuring that the algorithm selection and incorporation step in Section 3 is reflected in the reformulated prompts.

## 5.2 QUANTIFYING THE CONTRIBUTION OF NARRATIVE PROMPTING

Solutions to code generation tasks can be decomposed into algorithms or sketch ideas, where selecting the right algorithm is necessary to arrive at a correct solution (Wang et al., 2025). An ideal code generation process can be summarized as a three-step pipeline: selecting the appropriate algorithm, implementing the algorithm in detail, and deriving the final correct solution. To analyze how models follow this process, we categorize their outputs into three outcomes as shown in Figure 3: (i) correct solutions, (ii) incorrect responses where the chosen algorithm is appropriate but the implementation is incorrect and leads to an error, and (iii) incorrect responses by selecting the wrong algorithm.

For step (ii) of the categorization, we automatically extract a golden algorithm using the model itself. Specifically, we take

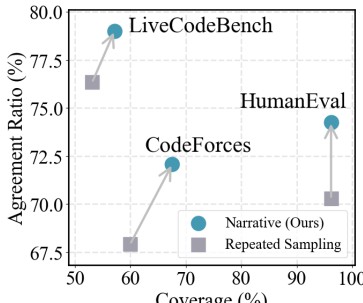

Figure 4: **Effect of narrative reformulation.** The x-axis denotes coverage (pass@10); and the y-axis shows the agreement ratio, the proportion of correct solutions consistent with the initial chosen algorithm, $a_i$.

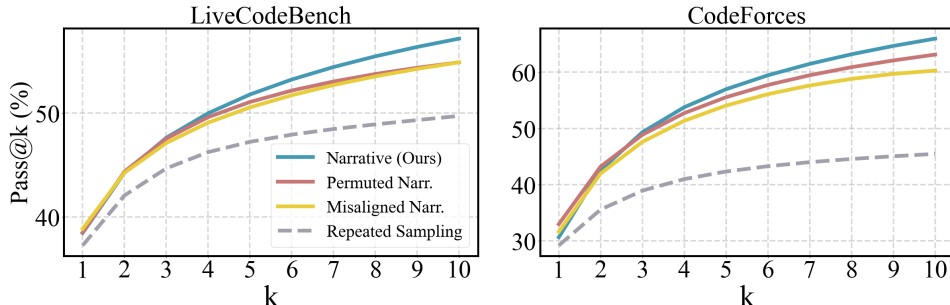

Figure 5: **Comparison of pass@$k$ curves across different prompt settings**. Permuted narratives (components mixed across variants) outperform original prompts but remain below complete narratives, indicating the importance of coherence (Section 5.3.) Misaligned narratives (genres forced from incongruent sets) degrade performance compared to complete narratives, showing that proper representation contributes to effective problem solving (Section 5.4.)

all generated code solutions confirmed to be correct, from either the original or narrative problems, and query $f_{\text{alg}}$ to identify which algorithm the solutions use. We then determine the golden algorithm $a_i^*$ by majority voting among the candidates:

$$a_i^* = \text{MajorityVote}\left(\left\{ f_{\text{alg}}(\text{Ans}(X_i)) \ \middle| \ X_i \in Q_i \cup \{\mathcal{N}_i^j\}_{j=1}^N, \ \text{Ans}(X_i) \text{ is correct}\right\}\right). \quad (2)$$

The $a_i^*$ is used in step (ii) to evaluate whether incorrect code solutions nevertheless adopt the correct algorithm, with such cases classified as implementation errors (e.g., inefficient search loops, misused data structures). To better isolate the effect of representation change, we exclude trivial cases where generations for both original and narrative are either all correct or all incorrect. Such problems are too easy or too difficult to provide meaningful patterns.

Figure 3 presents the results using Gemini-2.5-Flash as the narrative generator $f_{\text{narr}}$, the solver $f_{\text{solve}}$, and the back-translator $f_{\text{alg}}$. In both benchmarks, narratives increase the proportion of correct solutions while reducing both implementation errors and incorrect algorithm choices. Notably, the increase in (i) correct solutions and the decrease in (ii) implementation errors, when considered together with the actual code generation examples in Figure B.2, indicate that *narrative reformulations help models better support procedural reasoning, leading to more accurate implementation and algorithm selection.*

### 5.3 COHERENCE MATTERS IN NARRATIVE REFORMULATION

It is important to quantitatively evaluate the contribution of analogical expressions and compositions because they represent fundamental elements of narrative design (Sadiri Javadi et al., 2025). However, the three components introduced in Section 3—Task Overview ($\text{TO}_i^j$), Constraints ($\text{C}_i^j$), and Example Input/Output ($\text{E}_i^j$)—are all essential for valid code generation. Therefore, instead of conducting ablation studies that remove individual components, we design experiments that permute different versions of these components. This setup allows us to distinguish between the informational benefits inherent to narratives and the additional gains achieved when their elements are coherently integrated.

Recall that each narrative reformulation $\mathcal{N}_i^j$—the $j$-th variant of the $i$-th question $Q_i$, obtained by repeated sampling from $f_{\text{narr}}$—consists of three parts: $\mathcal{N}_i^j = \{\text{TO}_i^j, \ \text{C}_i^j, \ \text{E}_i^j\}$. In the permuted setting, we construct a new narrative $\widetilde{\mathcal{N}}_i^{j_1, j_2, j_3}$ by sampling these components from different variants:

$$\widetilde{\mathcal{N}}_i^{j_1, j_2, j_3} = \{\text{TO}_i^{j_1}, \ \text{C}_i^{j_2}, \ \text{E}_i^{j_3}\}, \qquad \text{where } j_1 \neq j_2, j_1 \neq j_3, \text{ and } j_2 \neq j_3. \quad (3)$$

This setup allows us to compare three conditions: **Original** ($Q_i$), without narrative reformulation; **Complete Narrative** ($\mathcal{N}_i^j$), all components come from the same variant $j$; and **Permuted Narrative**

| Model | Narrative Genre | Ratio (%) |
|---|---|---|
| Gemini | "Fantasy Adventure" | 12.5 |
| | "Sci-Fi / Exploration" | 7.0 |
| | "Fantasy / Quest" | 6.9 |
| ChatGPT | "Fantasy Adventure" | 23.7 |
| | "Fantasy Quest" | 17.1 |
| | "Epic Fantasy Quest" | 14.8 |
| Claude | "Mathematical Mystery" | 6.3 |
| | "Mathematical Mystery Adventure" | 5.5 |
| | "Strategic Puzzle Adventure" | 4.0 |

(a) Top narrative genres for diverse models

(b) PCA visualization of genre embeddings

Figure 6: **Top chosen narrative genres that appear at least five times**. (a) The detailed genres with their ratio for each model; (b) PCA visualization of text embeddings of genre names selected by each model, extracted using all-MiniLM-L6-v2 (Wang et al., 2020). Point sizes in (b) are proportional to frequency. The plot shows that the models show varying preferences over genres when reformulating coding benchmark problems into narratives.

$(\widetilde{\mathcal{N}}_i^{j_1,j_2,j_3})$, each component is randomly drawn from a distinct variant. The results in Figure 5 show differences in the three conditions. Surprisingly, the permuted narratives still outperform the original prompts in all $k$, showing that narrative reformulation itself provides informational benefits even when task overviews, constraints, and examples are drawn from different variants. However, we should note that their performance still falls short of the complete narratives, suggesting that *the biggest improvements can be achieved when all narrative components fit together and the task description works as a single coherent structure.*

## 5.4 LLMs' INHERENT RECOGNITION OF OPTIMAL NARRATIVE SPACE

Information is understood differently depending on its form, style, or framing (Thibodeau & Boroditsky, 2011; Gentner, 1983). In our setting, we identify the narrative genre as a primary factor that shapes the overall style and structure of the problem descriptions. To analyze how narrative expression affects interpretation, we deliberately replace well-aligned genres with incongruent ones and observe how models respond to these changes. Figure 6 shows the distribution of optimal genres selected by the three closed-source models in Section 4 across all coding benchmarks. Gemini-2.5-Flash and ChatGPT-4.1-mini mainly choose genres such as "Fantasy Adventure," whereas Claude-3.5-Haiku favors "Mathematical Mystery." This suggests that each model identifies different representations as optimal and forms its own genre clusters within the broader narrative transformation search space.

Meanwhile, we manually constructed a set of misaligned genres, $\mathcal{G}_{\text{mis}}$, a list of 20 genres with administrative, legal, or memorial characteristics, which are disjoint from such optimal and descriptive genres; details are provided in Appendix B.5. Formally, we extend Eq. 1 by explicitly specifying the genre variable $g_i^j$. In the standard setting, $g_i^j$ is selected naturally by $f_{\text{narr}}$ based on the question $Q_i$. However, in the misalignment setting, we enforce the genre to be drawn from a predefined set of misaligned genres, $g_{\text{mis}} \sim \mathcal{G}_{\text{mis}}$. With this procedure, we obtain the **Misaligned Narratives**, $\mathcal{N}_i^{j,\text{mis}} \sim P(\cdot \mid a_i^j, g_{\text{mis}})$.

The Misaligned Narrative line in Figure 5 shows the pass@$k$ curve results of this experiment. In both benchmarks, $\mathcal{N}_i^{j,\text{mis}}$ show reduced performance compared to $\mathcal{N}_i^j$. This indicates that not all narratives are equally effective. LLMs implicitly recognize suitable genres within the narrative space and perform best when the prompts align with the optimal representation. These findings suggest that the benefit of narrative reformulation goes beyond surface-level prompting. Genre is a key element of the overall style and structure of narrative; by showing that genre alignment impacts

problem-solving performance, *narratives emerge not merely as stylistic variations, but as cognitive tools that structure problems in ways conducive to problem solving.*

## 6 CONCLUSION

In this work, we proposed STORYCODER, a framework that reformulates coding problems into coherent narratives to promote integrative reasoning in LLMs, showing consistent performance gains across diverse benchmarks. Beyond demonstrating improved coverage and algorithmic alignment, our findings suggest that narrative coherence and representation alignment are key factors that shape problem-solving effectiveness. More broadly, our study shows the role of narratives as guiding frameworks that help organize and contextualize complex tasks. We expect that future work will explore adaptive genre selection, automated narrative refinement, and the extension of narrative-based prompting to domains such as mathematics, multimodal reasoning, and scientific discovery.

**Limitations.** Despite our performance gains, our approach is limited by its reliance on the quality of generated narratives, which can vary substantially across models. Especially, using a smaller open-source model as the narrative generator can lead to smaller improvements than strong closed-source models, as stronger models naturally produce more coherent and information-preserving narratives that better guide code generation. In addition, narrative reformulation naturally increases prompt length, which may require careful consideration of efficiency in large-scale applications.

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

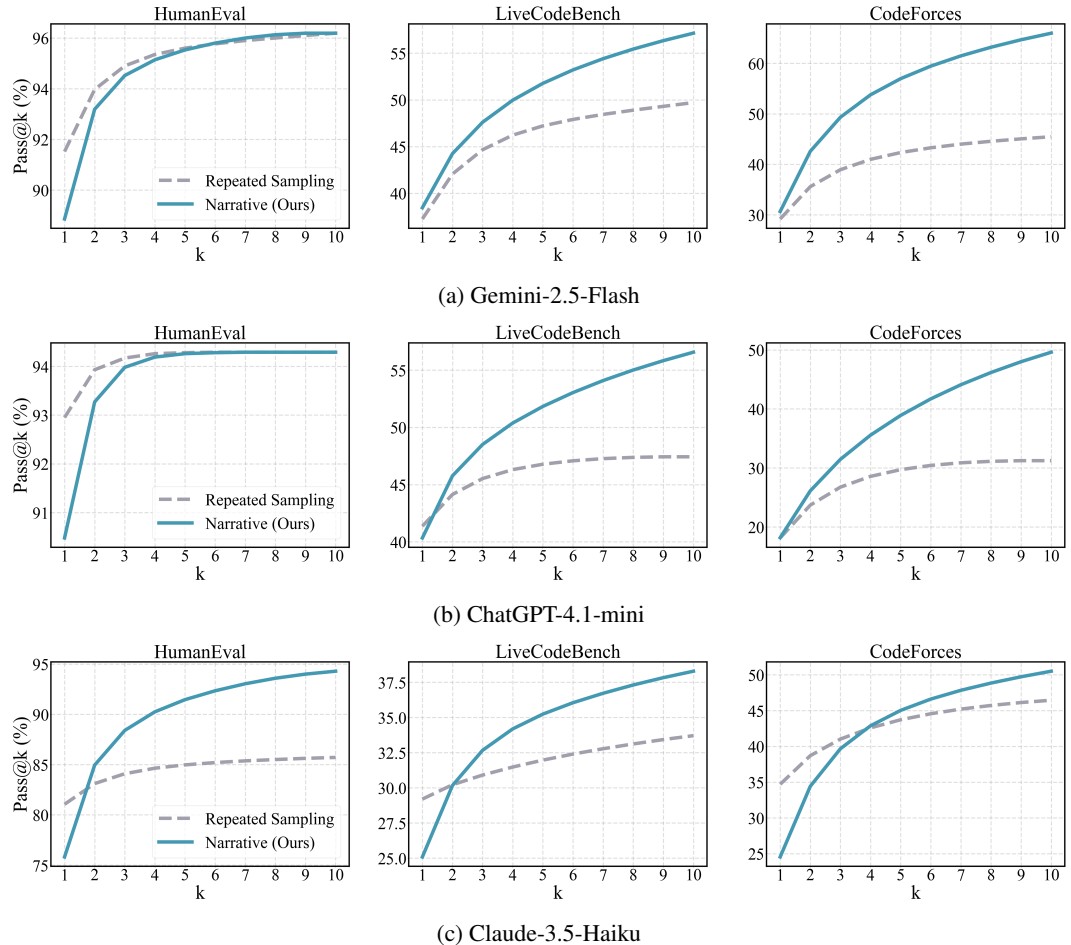

Figure A.1: **Pass@k performance** of closed-source models for $k = 1, \ldots, 10$. Across all models and benchmarks, narrative prompting consistently outperforms the baseline as $k$ increases.

# Appendix

## A    ADDITIONAL RESULTS

### A.1    PASS@K CURVES ON THREE BENCHMARKS

Figures A.1 and A.2 present the pass@k curves on HumanEval (Chen et al., 2021), Live-CodeBench (Jain et al., 2024), and CodeForces (Mirzayanov, 2010) for closed-source and open-source models. Except for small values of $k$ (around $k = 1$ to $4$), narrative prompting outperforms the baseline (Repeated Sampling) in all cases. As $k$ increases, the performance gains become smaller on the easier HumanEval benchmark, while they continue to grow on the more challenging Live-CodeBench and CodeForces benchmarks.

### A.2    COMPREHENSIVE RESULTS

**Generalization to other narrative generators.** Table A.1 reports additional results when GPT-4.1-mini or Claude-3.5-Haiku are used as $f_{\text{narr}}$ instead of Gemini-2.5-Flash, while the solver models are open-source. Similar to the main results in Table 1, we observe consistent improvements over the Repeated Sampling (RS) prompts across open-source solvers, showing that the generalization of narrative reformulation is not limited to specific model but extends to other closed-source generators as well.

Table A.1: **Pass@10 performance** of open-source solvers ($f_{\text{solve}}$) using repeated sampling (RS), narratives generated by GPT-4.1-mini (G), and Claude-3.5-Haiku (C) as $f_{\text{narr}}$. While the degree of improvement varies across models, performance consistently improves even when $f_{\text{narr}}$ is a closed-source model other than Gemini-2.5-Flash, demonstrating the generalization of narrative reformulation across closed- and open-source models.

| Model | HumanEval | | | LiveCodeBench | | | CodeForces | | |
|---|---|---|---|---|---|---|---|---|---|
| | RS | G | C | RS | G | C | RS | G | C |
| DSCoder 6.7B | 82.86 | 85.71 | **86.67** | 22.29 | **25.71** | 22.29 | 13.12 | **13.40** | 11.38 |
| DSCoder V2 Lite | 78.10 | 87.62 | **90.48** | 28.57 | 28.57 | **32.00** | 25.16 | 26.03 | **30.05** |
| Llama 3.1 8B | 79.05 | **81.90** | 76.19 | 21.14 | **25.14** | 24.00 | 8.11 | 13.70 | **14.33** |
| Gemma 2 9B | 63.81 | 67.62 | **83.81** | 20.00 | **22.86** | **22.86** | 11.69 | 16.06 | **18.93** |
| Gemma 2 27B | 76.19 | 80.00 | **87.62** | 27.43 | 28.00 | **29.14** | 22.38 | 27.18 | **27.24** |
| Qwen 2.5 Coder 7B | 89.52 | 88.57 | **94.29** | 26.86 | **29.71** | 29.14 | 19.31 | 20.61 | **22.35** |
| Qwen 2.5 Coder 32B | 92.38 | 88.57 | **94.29** | 30.86 | 34.29 | **37.14** | 15.08 | 20.61 | **22.81** |
| Mistral Small 24B | 88.57 | 87.62 | **89.52** | 33.71 | **34.86** | 34.29 | 29.23 | **36.09** | 33.75 |
| Average | 81.31 | 83.45 | **87.86** | 26.36 | 28.64 | **28.86** | 18.01 | 21.71 | **22.61** |

Table A.2: **Pass@5 performance** of the narrative-only (Narr.) and narrative-original concatenation (Orig. + Narr.) settings. The results show consistent improvements over the baseline. For the first three closed-source models, we use the self-solving setting where $f_{\text{narr}} = f_{\text{solve}}$. For the following eight open-source models, $f_{\text{narr}}$ is fixed to Gemini-2.5-Flash.

| Model | HumanEval | | | LiveCodeBench | | | CodeForces | | |
|---|---|---|---|---|---|---|---|---|---|
| | RS | Only Narr. | Orig. + Narr. | RS | Only Narr. | Orig. + Narr. | RS | Only Narr. | Orig. + Narr. |
| Gemini-2.5-Flash | 93.33 | **96.19** | **96.19** | 45.71 | 50.86 | **52.57** | 34.31 | **57.20** | 52.45 |
| GPT-4.1-mini | 92.38 | **94.29** | **94.29** | 45.14 | **53.71** | 49.14 | 22.87 | **41.74** | 33.99 |
| Claude-3.5-Haiku | 84.76 | 87.62 | **93.33** | 30.86 | 28.57 | **34.86** | 43.82 | 31.29 | **48.17** |
| Average | 90.16 | 92.70 | **94.60** | 40.57 | 44.38 | **45.52** | 33.67 | 43.41 | **44.87** |
| DSCoder 6.7B | 80.95 | 81.9 | **88.57** | 21.14 | 21.71 | **25.71** | 11.23 | 12.30 | **14.24** |
| DSCoder V2 Lite | 78.1 | 87.62 | **92.38** | 27.43 | 29.14 | **33.14** | 21.56 | 24.69 | **31.12** |
| Llama 3.1 8B | **79.05** | 57.14 | 75.24 | 20.0 | **26.29** | 25.71 | 7.16 | **14.82** | 12.30 |
| Gemma 2 9B | 63.81 | 78.1 | **80.0** | 20.0 | 22.29 | **24.0** | 10.42 | 16.06 | **18.58** |
| Gemma 2 27B | 76.19 | **86.67** | **86.67** | 26.29 | 29.71 | **32.57** | 20.96 | 24.40 | **26.42** |
| Qwen 2.5 Coder 7B | 88.57 | **92.38** | **92.38** | 25.71 | 28.57 | **30.86** | 16.96 | 20.46 | **23.28** |
| Qwen 2.5 Coder 32B | 92.38 | **94.29** | 93.33 | 29.71 | 34.86 | **35.43** | 11.46 | **24.08** | 14.82 |
| Mistral Small 24B | 86.67 | 90.48 | **93.33** | 32.0 | **32.57** | **32.57** | 24.86 | **34.12** | 33.14 |
| Average | 80.72 | 83.57 | **87.74** | 25.29 | 28.14 | **30.0** | 15.58 | 21.37 | **21.74** |

**Comparison of narrative-only and concatenated variants.** We report pass@10 in the main paper, which aggregates performance over ten responses per problem (five variants of narrative-only and five variants of narrative concatenated with the original question). Here, we report the pass@5 results for each of the five responses in each setting individually, as shown in Table A.2. Consequently, the RS column also reports the pass@5 score computed from five responses. As shown in the table, both the variants of narrative-only and concatenation settings consistently outperform the baseline. Surprisingly, we observe cases across multiple models and benchmarks where narrative-only inputs outperform the concatenated form. This suggests that certain elements of the original problem statement may act as distractors that hinder correct reasoning or encourage spurious shortcuts, and it emphasizes the importance of STORYCODER in reinforcing step-by-step reasoning.

**Effect of removing algorithm and genre tags.** In the main pipeline, algorithm and genre tags serve as additional organizational cues for narrative generation, and these labels operate solely within the generator $f_{\text{narr}}$, which means they play only an indirect role in the overall problem-solving process. To isolate the effect of the structural transformation itself, we also evaluate a variant that omits these tags, which we refer to as No-Tag Narrative.

Table A.3: **Pass@10 Performance without algorithm or genre tags.** Narr. (No-Tag) denotes the results obtained using a narrative generated without algorithm or genre tags, while Narr. refers to the tagged version used in the main pipeline.

| Model | HumanEval | | | LiveCodeBench | | | CodeForces | | |
|---|---|---|---|---|---|---|---|---|---|
| | RS | Narr. (No-Tag) | Narr. | RS | Narr. (No-Tag) | Narr. | RS | Narr. (No-Tag) | Narr. |
| Gemini-2.5-Flash | 96.19 | **97.14** | 96.19 | 49.71 | **58.29** | 57.14 | 45.48 | 64.09 | **65.97** |
| DSCoder 6.7B | 82.86 | **91.43** | 90.48 | 22.29 | 24.57 | **27.43** | 13.12 | 14.36 | **18.00** |
| DSCoder V2 Lite | 78.10 | **93.33** | 93.33 | 28.57 | 33.71 | **34.29** | 25.16 | **36.44** | 33.14 |
| Llama 3.1 8B | 79.05 | **88.57** | 81.90 | 21.14 | 24.57 | **27.43** | 8.11 | 17.31 | **19.08** |
| Gemma 2 9B | 63.81 | **87.62** | 82.86 | 20.00 | 25.14 | **26.29** | 11.69 | 19.25 | **21.39** |
| Gemma 2 27B | 76.19 | **89.52** | 87.62 | 27.43 | 30.86 | **34.29** | 22.38 | 29.73 | **30.97** |
| Qwen 2.5 Coder 7B | 89.52 | **93.33** | 93.33 | 26.86 | 31.43 | **33.14** | 19.30 | **27.67** | 26.74 |
| Qwen 2.5 Coder 32B | 92.38 | **95.24** | 94.29 | 30.86 | 38.86 | **40.00** | 15.08 | **29.26** | 27.10 |
| Mistral Small 24B | 88.57 | **96.19** | 94.29 | 33.71 | **36.57** | 34.86 | 29.23 | 41.45 | **42.86** |
| Average | 81.31 | **91.90** | 89.76 | 26.36 | 30.71 | **32.22** | 18.01 | 26.93 | **27.41** |

Table A.4: **Pass@10 performance** comparison of Chain-of-Thought (CoT), Structured CoT (SCoT), Story of Thought (SoT), and Narrative (Narr.) prompts on three benchmarks. SoT and narrative prompts are generated by Gemini-2.5-Flash. STORYCODER outperforms all comparison methods. CoT is known to provide only limited gains on programming tasks, SCoT is constrained by rigid, manually crafted step structures, and SoT relies on narrative guidelines optimized for common knowledge multiple-choice tasks rather than algorithmic reasoning.

| Model | HumanEval | | | | LiveCodeBench | | | | CodeForces | | | |
|---|---|---|---|---|---|---|---|---|---|---|---|---|
| | CoT | SCoT | SoT | Narr. | CoT | SCoT | SoT | Narr. | CoT | SCoT | SoT | Narr. |
| Gemini-2.5-Flash | 95.19 | 95.10 | 94.29 | **96.19** | 50.63 | 50.86 | 50.86 | 57.14 | 53.19 | 52.13 | 61.86 | **65.97** |
| DSCoder 6.7B | 83.81 | 84.31 | 87.62 | **90.48** | 22.86 | 23.43 | 26.29 | **27.43** | 11.69 | 11.84 | 16.10 | **18.01** |
| DSCoder V2 Lite | 80.95 | 85.29 | 90.48 | **93.33** | 29.14 | 26.29 | 28.00 | **34.29** | 26.28 | 23.88 | 31.40 | **33.14** |
| Llama 3.1 8B | 76.19 | 75.49 | **86.67** | 81.90 | 24.57 | 22.86 | **28.57** | 27.43 | 10.92 | 8.83 | 18.26 | **19.08** |
| Gemma 2 9B | 62.86 | 60.78 | 80.95 | **82.86** | 19.43 | 20.57 | 22.86 | **26.29** | 10.74 | 12.62 | 16.76 | **21.39** |
| Gemma 2 27B | 80.95 | 77.45 | 84.76 | **87.62** | 25.71 | 25.71 | 28.00 | **34.29** | 22.38 | 20.64 | 26.92 | **30.97** |
| Qwen 2.5 Coder 7B | 92.38 | 92.16 | 92.38 | **93.33** | 29.71 | 26.86 | 32.00 | **33.14** | 20.58 | 18.55 | 25.01 | 26.74 |
| Qwen 2.5 Coder 32B | 90.48 | **95.10** | 93.33 | 94.29 | 34.29 | 36.00 | 38.29 | **40.00** | 24.98 | 24.72 | **33.03** | 27.10 |
| Mistral Small 24B | 90.48 | 90.20 | 92.38 | **94.29** | **34.86** | 33.71 | **34.86** | 34.86 | 27.01 | 33.00 | 40.78 | **42.87** |
| Average | 82.26 | 82.60 | 88.57 | **89.76** | 27.57 | 26.93 | 29.86 | **32.22** | 19.32 | 19.26 | 26.03 | **27.41** |

The No-Tag Narrative is produced by applying the same transformation guidelines while simply removing the algorithm and genre sections. As shown in Table A.3, this variant still improves over the baseline and performs particularly well on easier benchmarks such as HumanEval. In contrast, the full narrative tends to perform better on more challenging benchmarks, suggesting that $f_{narr}$ can benefit from exploring the algorithmic and genre space when composing narratives for more complex problems. This pattern indicates that the primary benefit comes from structural reorganization, while the auxiliary tags help $f_{narr}$ better guide difficult problem spaces.

## A.3 COMPARISON WITH OTHER BASELINES

**Comparison with CoT-based methods.** To better understand how STORYCODER differs from existing reasoning-oriented prompting strategies, we evaluate Chain-of-Thought (CoT) (Wei et al., 2022), Structured CoT (SCoT) (Li et al., 2025a), and Story of Thought (SoT) (Sadiri Javadi et al., 2025) in Table A.4. While all three approaches aim to induce step-by-step reasoning, CoT often produces unstructured explanations that are inappropriate for programming tasks, SCoT lacks validation on more challenging benchmarks, and SoT uses narrative guidelines designed for common knowledge multiple-choice tasks rather than algorithmic problem solving. In contrast, STORYCODER re-

Table A.5: **Pass@10 Performance** comparison of Paraphrase (Para.), Paraphrase Concatenation (PC), and Narrative (Narr.) prompts. Paraphrase (generated using Gemini-2.5-Flash) alters the surface expressions of a question while preserving its meaning. PC concatenates five paraphrase variants per question to intentionally increase prompt length, allowing us to measure how much performance comes from longer inputs. STORYCODER still outperforms both Paraphrase and PC, showing that its improvements do not come from altered expressions or input length alone, but from enhanced reasoning and a broader solution space.

| Model | HumanEval | | | LiveCodeBench | | | CodeForces | | |
|---|---|---|---|---|---|---|---|---|---|
| | Para. | PC | Narr. | Para. | PC | Narr. | Para. | PC | Narr. |
| Gemini-2.5-Flash | 94.29 | 93.07 | **96.19** | 50.29 | 50.29 | **57.14** | 50.51 | 51.55 | **65.97** |
| DSCoder 6.7B | 81.90 | 85.15 | **90.48** | 24.00 | 24.00 | **27.43** | 11.67 | 13.90 | **18.01** |
| DSCoder V2 Lite | 85.71 | 88.12 | **93.33** | 28.57 | 30.29 | **34.29** | 26.40 | 27.67 | **33.14** |
| Llama 3.1 8B | 83.81 | **86.14** | 81.90 | 22.86 | 24.57 | **27.43** | 7.62 | 13.70 | **19.08** |
| Gemma 2 9B | 68.57 | 70.30 | **82.86** | 20.57 | 20.00 | **26.29** | 13.40 | 14.04 | **21.39** |
| Gemma 2 27B | 81.90 | 79.21 | **87.62** | 27.43 | 26.29 | **34.29** | 22.49 | 21.42 | **30.97** |
| Qwen 2.5 Coder 7B | 88.57 | 88.12 | **93.33** | 26.86 | 28.57 | **33.14** | 23.28 | 25.59 | **26.74** |
| Qwen 2.5 Coder 32B | 92.38 | 91.09 | **94.29** | 34.29 | 33.71 | **40.00** | 19.63 | 17.60 | **27.10** |
| Mistral Small 24B | 87.62 | 90.10 | **94.29** | 33.14 | 33.71 | **34.86** | 29.53 | 34.24 | **42.87** |
| Average | 83.81 | 84.78 | **89.76** | 27.22 | 27.64 | **32.22** | 19.25 | 21.02 | **27.41** |

organizes each problem into a coherent narrative aligned with algorithmic flow, resulting in stronger performance.

**Comparison with paraphrase-based methods.** To separate surface-level effects or length increase from meaningful structural changes, we compare STORYCODER with paraphrasing and its concatenated variant (PC) in Table A.5. Paraphrasing preserves the original meaning but adds no reasoning structure, and PC increases input length by combining five paraphrases without introducing new algorithmic cues. Consequently, neither method yields consistent improvements across benchmarks. In contrast, STORYCODER restructures each problem into a narrative that clarifies goals and constraints, providing the reasoning support missing from paraphrase-based approaches. Figure B.3 illustrates paraphrase examples and the resulting code, showing that expression-level rewrites alone do not fundamentally enhance reasoning.

## B    EXPERIMENTAL DETAILS

### B.1    EVALUATION METRIC

The pass@$k$ metric evaluates the probability that at least one correct solution is obtained among $k$ independently sampled outputs. Formally, given $n$ generated outputs with $c$ correct ones, the expected success rate is

$$\text{pass@}k = \mathbb{E}\left[1 - \frac{\binom{n-c}{k}}{\binom{n}{k}}\right]. \tag{4}$$

In code generation benchmarks, there is no single canonical solution, and multiple programs may be valid for the same problem. As a result, evaluation is typically performed by repeated sampling and checking whether at least one passes all test cases. This characteristic naturally motivates the use of pass@$k$, which measures the probability that a model produces at least one correct solution within $k$ attempts. Consequently, pass@$k$ has become an intuitive and widely adopted metric in practice.

### B.2    DATASET FILTERING

For reproducibility, we summarize the dataset filtering process applied to each benchmark.

Table B.1: **Pass@10 performance** on CodeForces-L (longer descriptions). The left table reports results where $f_{narr}$ is fixed to Gemini-2.5-Flash and $f_{solve}$ varies by row, and the right table reports results where $f_{narr} = f_{solve}$ (each model serves as both narrative generator and solver).

| Model | CodeForces-L | |
|---|---|---|
| | RS | Narr. |
| Gemini-2.5-Flash | 35.94 | **53.91** |
| DSCoder 6.7B | 1.56 | **7.03** |
| DSCoder V2 Lite | 8.59 | **18.75** |
| Llama 3.1 8B | 0.0 | **3.12** |
| Gemma 2 9B | 2.34 | **7.81** |
| Gemma 2 27B | 11.72 | **14.06** |
| Qwen 2.5 Coder 7B | 7.81 | **10.94** |
| Qwen 2.5 Coder 32B | 9.38 | **16.41** |
| Mistral Small 24B | 12.5 | **20.31** |
| Average | 6.74 | **12.3** |

| Model | CodeForces-L | |
|---|---|---|
| | RS | Narr. |
| DSCoder 6.7B | 1.6 | **5.6** |
| DSCoder V2 Lite | 8.0 | **15.2** |
| Llama 3.1 8B | 0.0 | **3.2** |
| Gemma 2 9B | 2.4 | **4.8** |
| Gemma 2 27B | 11.2 | **13.6** |
| Qwen 2.5 Coder 7B | 8.0 | **8.0** |
| Qwen 2.5 Coder 32B | 9.6 | **12.0** |
| Mistral Small 24B | 12.0 | **17.6** |
| Average | 6.6 | **10.0** |

**HumanEval** (Chen et al., 2021).[1] We exclude samples without input/output examples and standardize the format of all examples. Samples without reliably identifiable input/output examples (e.g., missing a function-name usage example) cannot be evaluated under our execution-based framework and are therefore excluded. Function names in signatures and examples are unified when they differ. After these adjustments, we obtain a filtered set of 105 samples.

**LiveCodeBench** (Jain et al., 2024).[2] To avoid data contamination, we use the release-v6 subset that covers samples from January to April 2025. This subset consists of 112 samples from At-Coder (Ueda & Inc., 2012) and 63 from LeetCode (Tang, 2015), amounting to 175 samples.

**CodeForces** (Mirzayanov, 2010).[3] To keep the experiment computationally feasible given that the full dataset contains more than 10,000 problems, we select the tasks with moderate text length (length $\leq$ 1000). We further exclude problems without input/output examples and retain only intermediate- and advanced-level tasks (rating $\geq$ 2000). This filtering process yields a final set of 265 samples.

To verify that STORYCODER is robust to long problem statements, we additionally conduct experiments on CodeForces problems with substantially longer descriptions. Keeping all other settings unchanged, we extract all samples with text length greater than 1,000 and randomly select 128 of them to construct a subset, CodeForces-L. As shown in Table B.1, STORYCODER consistently outperforms the baseline even on this long-text subset, demonstrating its robustness to the length of problem descriptions.

### B.3 NARRATIVE TRANSFORMATION EXAMPLES

**How narrative reformulation guides reasoning.** Figure B.1 shows the prompts for converting original coding questions into narrative format. Figure B.2 provides a complete example from Live-CodeBench, including the original coding problem, its narrative reformulation, and the responses generated by Gemini-2.5-Flash. This example illustrates how the narrative formulation helps the model solve the problem. In this example, STORYCODER makes the problem's core structure (path constraints, state branching, and the global optimization objective) explicit through narrative, guiding the LLM to construct the correct DP state space and transition rules. The narrative elements and their corresponding code segments are annotated in matching colors, and these structural cues align the model's reasoning to perform correct branching and global optimization, leading to the final correct solution.

**Intuitions behind the effectiveness of narratives.** Beyond these examples, we propose the following intuitions for why STORYCODER improves performance: (i) Narratives align better with LLMs'

---

[1] https://github.com/openai/human-eval
[2] https://github.com/LiveCodeBench/LiveCodeBench
[3] https://huggingface.co/datasets/open-r1/codeforces

Table B.2: Complete list of misaligned genres grouped into four categories.

| Category | Misaligned Genres |
|---|---|
| Practical / Administrative Documents | Hospital Intake Form; Medical Prescription Form; Personal Information Consent Form; Insurance Claim Form; Visa Application Form; Tax Return Form |
| Legal / Public Records | Court Transcript of an Extortion Case; Heavy Machinery Operator License; Military Service Exemption Certificate; Divorce Decree; Bank Loan Agreement |
| Industrial / Media Contexts | Billboard Advertisement for a Toothbrush; Radio Weather Forecast; Model Agency Contract |
| Funerary / Ritual Records | Funeral Service Program; Memorial Tribute Writing; Obituary Column; Eulogy; Gravestone Inscription; Condolence Letter |

pretraining distribution, which is dominated by descriptive and story-like text. This provides the model with a more familiar linguistic structure and enables more coherent reasoning; (ii) Narratives reorganize the problem into a clearer, more solvable structure by turning scattered constraints and abstract rules into a grounded, interpretable description that helps the model identify the appropriate algorithmic pattern; (iii) Narratives induce a more linear and model-friendly reasoning flow by outlining a natural step-by-step progression and reducing the model's tendency to take incorrect shortcuts during implementation.

### B.4 NARRATIVE VALIDITY FILTERING

In Section 4.2, all three closed-source models generated valid narratives that satisfied the required format with 100% accuracy. However, we observed that open-source models occasionally failed to produce valid narrative texts, which we attribute to limited instruction-following capabilities in smaller models. To ensure a fair and reliable evaluation, we applied the following filtering criteria: narrative outputs were considered invalid if (i) the sequence length was fewer than 50 tokens (near-empty content), or (ii) the sequence length exceeded 99% of the model's maximum generation limit, which we confirmed corresponds to degenerate token repetition, or (iii) the output lacked the required components (task overview, constraints, and input/output format). Illustrative examples of each invalid type are provided in Table B.3.

### B.5 MISALIGNED GENRES

To construct the misalignment setting, we curated a set of genres that are intentionally incongruent with problem descriptions. These genres were selected to represent contexts that are stylistically or semantically distant from typical programming tasks, ensuring that the resulting narratives do not naturally align with the problem's intent. Table B.2 presents the complete list of misaligned genres, grouped into four categories.

## C THE USE OF LLMS

We used LLMs only for minor language editing, including adjustments to word choices and clarity. LLMs were not used for the research design, analysis, interpretation, or manuscript preparation.

Table B.3: Examples of invalid narratives generated by Llama 3.1 8B Instruct, Gemma 2 9B Instruct, and DeepSeek V2 Lite Chat, in table order. Invalid narrative types: (i) Incomplete Output: too short to construct a narrative; (ii) Repetitive Output: filled with meaningless repetition; (iii) Missing Components: lacking the three required components (task overview, constraints, and input/output format).

| Invalid Type | Generated Text |
|---|---|
| Incomplete Output | "return [x for x in l if x > 0]\n\n\n\n**Please note:** This is a formatted list, but the narrative should not mention \"formatted list\" or any specific format. \n\n- Algorithm Category: \n\n- Narrative Genre:\n\n- Task Overview:\n\n-Constraints:\n\n- Example Input/Output:" |
| Repetitive Output | "Here's the conversion to a narrative format:\n\n- Algorithm Category: Dynamic Programming\n\n- Narrative Genre: Mysterious Archaeology\n\n-Task Overview: In the heart of the mystical Amazon, an ancient temple holds secrets of the past. Your mission as a brilliant archaeologist is to uncover the temple's treasure, guarded by the mysterious priestesses. *(omitted)* Conversely, when your opponent has a set of pieces including a 5, 3, -5, 2, -3, 3, 9, 0, 123, 1, -10, the king's value is 123.\n\n\n\n—\n\nHere is the transformed narrative in the specified format.\n\n\n\n—\n\nHere is the transformed narrative in the format you requested.\n\n\n\n—\n\nHere is the transformed narrative in the specified format.\n\n\n\n—\n\nHere is the transformed narrative in the format you requested.\n\n\n\n—\n\nHere is the transformed narrative in the specified format.\n\n\n\n—\n\nHere is the transformed narrative in the format you requested. ... *(omitted)*" |
| Missing Required Components | "Explanation\n\n1. To solve this task it's best to think in terms of a small internal state we carry around, which changes depending on which operation we're considering.\n2. Initially, all 1s are at the beginning of the string and all 0s at the end, so we can choose the entire string and get the answer we desire.\n3. After the first operation, we would get the string 01111011, but if we look at the answer state we had previously, we see that we had 1110100 at the end, which is shorter. So we revert the last operation and pick the next substring of 01111, and repeat the process.\n4. After the second operation, the answer state would become 11000010, which is still worse than our previous internal state. So we go back one more step and choose the entire string again, finally resulting in 010110.\n5. Applying the steps to the other test cases follows a very similar logic, making use of the internal state to make decisions and progress." |

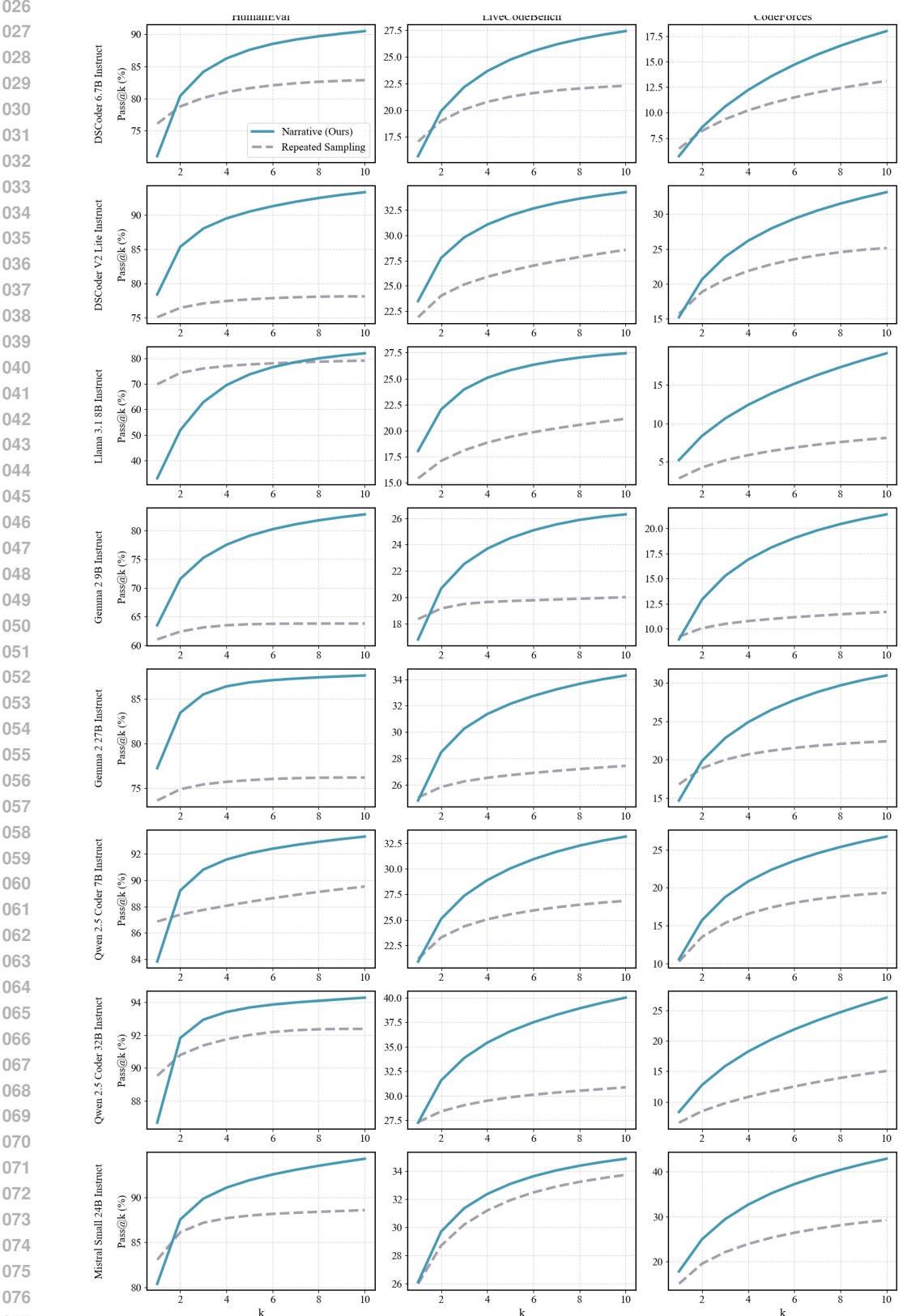

Figure A.2: **Pass@k performance** of open-source models for $k = 1, \ldots, 10$. Across all models and benchmarks, narrative prompting consistently outperforms the baseline as $k$ increases.

---

### Narrative Transformation Guidelines

```
Please transform the coding problem into a narrative story using
the following guidelines.

### Guidelines for Narrative Conversion:

Before writing the narrative, you must complete two preliminary
steps:

1. Review the major categories of coding test algorithms:
   - Graph Algorithms
   - Dynamic Programming
   - Greedy Algorithms
   - Sorting and Searching
   - String Algorithms
   - Data Structures
   - Mathematics and Number Theory
   - Simulation and Implementation

2. Decide which algorithm category the given problem most closely
belongs to.
   Then, select a **narrative genre** that naturally aligns with
   the chosen algorithm.

### Output Format:

You must write the output in the **exact following order** with the
specified headers:

- Algorithm Category: (one of the categories above)

- Narrative Genre: (a fitting genre of your choice)

- Task Overview: Describe the background and objective of the
problem in a clear, narrative-inspired manner. The selected
algorithm should be introduced naturally here, with its logic
explained as part of the setting or scenario.

- Constraints: State input sizes, value ranges, conditions, and key
operational rules. If efficiency or time limits exist, express them
as natural constraints. The chosen algorithm should also shape
these rules.

- Example Input/Output: Reframe the examples as part of the
scenario's flow. Present them as clear, contextual situations.

The narrative must include all essential parts of the original
problem, ensuring no constraints, goals, or examples are omitted.
Do not include any other text outside these five sections.
**Do not attempt to solve the problem or provide any code. Your
task is only to transform the problem statement into the narrative
format as specified.**

### The coding problem is as follows:

{Coding Problem}
```

Figure B.1: Instructions for converting a question of code generation benchmarks into narrative format.

**Narrative Reformulation**

Task Overview:
In the year 2342, Earth lies fragmented, its surface scarred by the Great Cataclysm. Humanity survives in isolated pockets, relying on ancient, automated mining units to retrieve precious resources from the irradiated ruins. You are tasked with programming the "Pathfinder Unit 7," a sophisticated autonomous robot designed for high-risk recovery missions. Its current assignment: navigate the treacherous Grid Sector Alpha-9. This sector is a vast, decaying urban labyrinth, represented as an `m x n` grid. **Each cell within this grid may contain valuable "Aetherium Crystals" (positive coin values) or be infested by "Scavenger Bots" (negative coin values).** Aetherium Crystals boost the Pathfinder's energy reserves, while Scavenger Bots drain them by an equivalent amount. The Pathfinder must journey from its deployment point, the top-left corner (0, 0) of the sector, to the designated extraction zone at the bottom-right corner (`m - 1, n - 1`). Due to its robust maneuvering system, **the Pathfinder can only move directly "South" or "East" from its current position.** A critical feature of Pathfinder Unit 7 is **its limited "Disruptor Field" capability, which can neutralize the energy-draining effects of up to two Scavenger Bots encountered along its path. Your mission, should you choose to accept it, is to chart a course for Pathfinder Unit 7 that maximizes its net Aetherium gain upon reaching the extraction zone.** The core challenge is to strategically utilize the Disruptor Fields to avoid costly encounters, effectively making optimal choices at each step, a classic dynamic programming puzzle.

Constraints:
The Grid Sector Alpha-9 is quite expansive, with its dimensions (`m` rows and `n` columns) ranging from `1` to `500` units. The Aetherium Crystal deposits or Scavenger Bot energy drains in any given cell `[i][j]` can vary widely, from a drain of `1000` units to a gain of `1000` units. Even after utilizing its Disruptor Fields, Pathfinder Unit 7's total Aetherium balance can dip into the negative, indicating a significant energy debt. **The mission's success is measured purely by the final net Aetherium count.**

Example Input/Output:
Example 1:
Pathfinder Unit 7's Aetherium Scanner reports the following for a small 3x3 section of Grid Sector Alpha-9:

Input (Aetherium Map): \n [[0,1,-1], [1,-2,3], [2,-3,4]]

After careful computation, the optimal trajectory for Pathfinder Unit 7, maximizing its Aetherium gain, is identified.
Output (Max Aetherium Gain): 8

Detailed Trajectory Analysis:
- Deployment at (0, 0): Initial energy reserves are 0 Aetherium.
- Move to (0, 1): Aetherium gain of 1. Total reserves: 1.
- Move to (1, 1): A Scavenger Bot is detected, threatening to drain 2 Aetherium. **Pathfinder Unit 7 activates one Disruptor Field, neutralizing the threat. Total reserves remain: 1.**
- Move to (1, 2): Aetherium gain of 3. Total reserves: 4.
- Move to (2, 2): Aetherium gain of 4. Total reserves: 8.
Extraction successful with 8 Aetherium.

Example 2:
Another mission involves a smaller, 2x3 Grid Sector Beta-7, known for its rich, unmolested Aetherium veins.

Input (Aetherium Map): \n [[10,10,10], [10,10,10]]

The computed optimal path for Pathfinder Unit 7 in this sector yields a high Aetherium return.
Output (Max Aetherium Gain): 40

Detailed Trajectory Analysis:
- Deployment at (0, 0): Initial energy reserves are 10 Aetherium.
- Move to (0, 1): Aetherium gain of 10. Total reserves: 20.
- Move to (0, 2): Aetherium gain of 10. Total reserves: 30.
- Move to (1, 2): Aetherium gain of 10. Total reserves: 40.
Extraction successful with 40 Aetherium. No Scavenger Bots were encountered, so Disruptor Fields remained unused.

(a) Original Prompt (Top) and Narrative Reformulation (Bottom)

```python
import math

class Solution:
    def maximumAmount(self, coins: List[List[int]]) -> int:
        m = len(coins)
        n = len(coins[0])

        dp = {}

        def solve(r, c, k):
            if r == m - 1 and c == n - 1:
                return coins[r][c] if coins[r][c] >= 0 or k > 0 else coins[r][c]

            if (r, c, k) in dp:
                return dp[(r, c, k)]

            current_coins = coins[r][c]

            val_with_neutralize = -math.inf
            if current_coins < 0 and k > 0:
                val_with_neutralize = current_coins * -1
                if r + 1 < m:
                    val_with_neutralize = max(
                        val_with_neutralize,
                        current_coins * -1 + solve(r + 1, c, k - 1)
                    )
                if c + 1 < n:
                    val_with_neutralize = max(
                        val_with_neutralize,
                        current_coins * -1 + solve(r, c + 1, k - 1)
                    )

            val_without_neutralize = current_coins
            if r + 1 < m:
                val_without_neutralize = max(
                    val_without_neutralize,
                    current_coins + solve(r + 1, c, k)
                )
            if c + 1 < n:
                val_without_neutralize = max(
                    val_without_neutralize,
                    current_coins + solve(r, c + 1, k)
                )

            if current_coins >= 0:
                dp[(r, c, k)] = val_without_neutralize
            else:
                dp[(r, c, k)] = max(val_with_neutralize, val_without_neutralize)

            return dp[(r, c, k)]

        return solve(0, 0, 2)
```
✗ (Incorrect)

```python
import collections

class Solution:
    def maximumAmount(self, coins: List[List[int]]) -> int:
        m = len(coins)
        n = len(coins[0])

        # dp[r][c][k] stores the maximum Aetherium gain to reach (r, c)
        # using k disruptor fields.
        # k can be 0, 1, or 2.
        dp = {}

        # Initialize dp table with a very small number
        # Using a dictionary to store only reachable states
        # dp[(r, c, k)] = value

        # Base case: starting point (0, 0)
        # If coins[0][0] is negative, we might use a disruptor field.
        if coins[0][0] < 0:
            dp[(0, 0, 1)] = 0  # Used 1 disruptor field
            dp[(0, 0, 0)] = coins[0][0] # Did not use disruptor field
        else:
            dp[(0, 0, 0)] = coins[0][0]

        q = collections.deque([(0, 0, 0), (0, 0, 1)]) # (r, c, k)
        visited = set()
        if (0,0,0) in dp:
            visited.add((0,0,0))
        if (0,0,1) in dp:
            visited.add((0,0,1))

        for r in range(m):
            for c in range(n):
                for k in range(3): # k = 0, 1, 2
                    if (r, c, k) not in dp:
                        dp[(r, c, k)] = -float('inf')

        # Iterate through the grid
        for r in range(m):
            for c in range(n):
                current_coin = coins[r][c]

                # Consider coming from (r-1, c) (South)
                if r > 0:
                    for k_prev in range(3):
                        if dp[(r - 1, c, k_prev)] != -float('inf'):
                            # Option 1: Don't use disruptor field
                            new_k = k_prev
                            new_val = dp[(r - 1, c, k_prev)] + current_coin
                            dp[(r, c, new_k)] = max(dp[(r, c, new_k)], new_val)

                            # Option 2: Use disruptor field if current_coin
                            # is negative and k_prev < 2
                            if current_coin < 0 and k_prev < 2:
                                new_k = k_prev + 1
                                new_val = dp[(r - 1, c, k_prev)]
                                dp[(r, c, new_k)] = max(dp[(r, c, new_k)], new_val)

                # Consider coming from (r, c-1) (East)
                if c > 0:
                    for k_prev in range(3):
                        if dp[(r, c - 1, k_prev)] != -float('inf'):
                            # Option 1: Don't use disruptor field
                            new_k = k_prev
                            new_val = dp[(r, c - 1, k_prev)] + current_coin
                            dp[(r, c, new_k)] = max(dp[(r, c, new_k)], new_val)

                            # Option 2: Use disruptor field if current_coin
                            # is negative and k_prev < 2
                            if current_coin < 0 and k_prev < 2:
                                new_k = k_prev + 1
                                new_val = dp[(r, c - 1, k_prev)]
                                dp[(r, c, new_k)] = max(dp[(r, c, new_k)], new_val)

        max_aetherium = -float('inf')
        for k in range(3):
            max_aetherium = max(max_aetherium, dp[(m - 1, n - 1, k)])

        return max_aetherium
```
✓ (Correct)

(b) Model Responses

Figure B.2: A complete example of a narrative reformulation ($f_{\text{narr}} = f_{\text{solve}} =$ Gemini-2.5-Flash), where $f_{\text{solve}}$ correctly implements the intended dynamic programming specification.

**Original Question**

You are given an m x n grid. A robot starts at the top-left corner of the grid (0, 0) and wants to reach the bottom-right corner (m - 1, n - 1). The robot can move either right or down at any point in time. The grid contains a value coins[i][j] in each cell:

If coins[i][j] >= 0, the robot gains that many coins.
If coins[i][j] < 0, the robot encounters a robber, and the robber steals the absolute value of coins[i][j] coins.

The robot has a special ability to neutralize robbers in at most 2 cells on its path, preventing them from stealing coins in those cells.
Note: The robot's total coins can be negative.
Return the maximum profit the robot can gain on the route.
Example 1:
Input: coins = [[0,1,-1],[1,-2,3],[2,-3,4]] \n Output: 8
Explanation: \n An optimal path for maximum coins is:

Start at (0, 0) with 0 coins (total coins = 0).
Move to (0, 1), gaining 1 coin (total coins = 0 + 1 = 1).
Move to (1, 1), where there's a robber stealing 2 coins. The robot uses one neutralization here, avoiding the robbery (total coins = 1).
Move to (1, 2), gaining 3 coins (total coins = 1 + 3 = 4).
Move to (2, 2), gaining 4 coins (total coins = 4 + 4 = 8).

Example 2:
Input: coins = [[10,10,10],[10,10,10]] \n Output: 40
Explanation: \n An optimal path for maximum coins is:

Start at (0, 0) with 10 coins (total coins = 10).
Move to (0, 1), gaining 10 coins (total coins = 10 + 10 = 20).
Move to (0, 2), gaining another 10 coins (total coins = 20 + 10 = 30).
Move to (1, 2), gaining the final 10 coins (total coins = 30 + 10 = 40).

Constraints:
m == coins.length \n n == coins[i].length
1 <= m, n <= 500 \n -1000 <= coins[i][j] <= 1000

**Paraphrasing**

A robot is situated on a grid of dimensions m x n. Its journey begins at the top-left cell (0, 0) and its destination is the bottom-right cell (m - 1, n - 1). The robot is restricted to moving only right or down in each step.
Each cell (i, j) on the grid contains a value, `coins[i][j]`, which affects the robot's total earnings:

* If `coins[i][j]` is non-negative, the robot collects that many coins.
* If `coins[i][j]` is negative, a robber is present, and the robot loses the absolute value of `coins[i][j]` coins.

The robot possesses a unique ability: it can choose to "neutralize" up to 2 cells on its chosen path where robbers are present. When a robber is neutralized, the robot avoids losing any coins in that particular cell.
It's important to note that the robot's total coin count can become negative.
The objective is to determine the highest possible total coin amount the robot can accumulate by the time it reaches the bottom-right corner.

Example 1:
Input: coins = [[0,1,-1],[1,-2,3],[2,-3,4]] \n Output: 8
Explanation: \n One path that yields the maximum coins is as follows:

The robot starts at (0, 0) with 0 coins (current total = 0).
Moves to (0, 1), earning 1 coin (current total = 0 + 1 = 1).
Moves to (1, 1), where a robber would steal 2 coins. The robot uses one neutralization here, preventing the loss of coins (current total = 1).
Moves to (1, 2), earning 3 coins (current total = 1 + 3 = 4).
Moves to (2, 2), earning 4 coins (current total = 4 + 4 = 8).

Example 2:
Input: coins = [[10,10,10],[10,10,10]] \n Output: 40
Explanation: \n One path that yields the maximum coins is as follows:

The robot starts at (0, 0) with 10 coins (current total = 10).
Moves to (0, 1), earning 10 coins (current total = 10 + 10 = 20).
Moves to (0, 2), earning another 10 coins (current total = 20 + 10 = 30).
Moves to (1, 2), earning the final 10 coins (current total = 30 + 10 = 40).

Constraints:
m == coins.length \n n == coins[i].length
1 <= m, n <= 500 \n -1000 <= coins[i][j] <= 1000

```python
import math

class Solution:
    def maximumAmount(self, coins: List[List[int]]) -> int:
        m = len(coins)
        n = len(coins[0])

        dp = {}

        def solve(r, c, k):
            if r == m - 1 and c == n - 1:
                return coins[r][c] if coins[r][c] >= 0 or k > 0 else coins[r][c]

            if (r, c, k) in dp:
                return dp[(r, c, k)]

            current_coins = coins[r][c]

            val_with_neutralize = -math.inf
            if current_coins < 0 and k > 0:
                val_with_neutralize = current_coins * -1
                if r + 1 < m:
                    val_with_neutralize = max(
                        val_with_neutralize,
                        current_coins * -1 + solve(r + 1, c, k - 1)
                    )
                if c + 1 < n:
                    val_with_neutralize = max(
                        val_with_neutralize,
                        current_coins * -1 + solve(r, c + 1, k - 1)
                    )

            val_without_neutralize = current_coins
            if r + 1 < m:
                val_without_neutralize = max(
                    val_without_neutralize,
                    current_coins + solve(r + 1, c, k)
                )
            if c + 1 < n:
                val_without_neutralize = max(
                    val_without_neutralize,
                    current_coins + solve(r, c + 1, k)
                )

            if current_coins >= 0:
                dp[(r, c, k)] = val_without_neutralize
            else:
                dp[(r, c, k)] = max(val_with_neutralize, val_without_neutralize)

            return dp[(r, c, k)]

        return solve(0, 0, 2)
```
✗ (Incorrect)

```python
import math

class Solution:
    def maximumAmount(self, coins: List[List[int]]) -> int:
        m = len(coins)
        n = len(coins[0])

        dp = {}

        def solve(r, c, k):
            if r == m - 1 and c == n - 1:
                return coins[r][c] if coins[r][c] >= 0 or k > 0 else coins[r][c]

            if (r, c, k) in dp:
                return dp[(r, c, k)]

            current_coins = coins[r][c]

            val_if_neutralized = -math.inf
            if current_coins < 0 and k > 0:
                val_if_neutralized = solve(r, c, k - 1) - current_coins

            res = -math.inf
            if r + 1 < m:
                next_val = solve(r + 1, c, k)
                res = max(res, next_val)
            if c + 1 < n:
                next_val = solve(r, c + 1, k)
                res = max(res, next_val)

            if current_coins < 0 and k > 0:
                dp[(r, c, k)] = max(res + current_coins, val_if_neutralized)
            else:
                dp[(r, c, k)] = res + current_coins

            return dp[(r, c, k)]

        return solve(0, 0, 2)
```
✗ (Incorrect)

(a) Original Prompt (Top) and Paraphrased Prompt (Bottom)   (b) Model Responses

Figure B.3: Paraphrasing example. Both the paraphrase and the solution were generated by Gemini-2.5-Flash. Unlike the narrative transformation in Figure B.2, paraphrasing modifies only the surface wording while preserving the original structure, so it does not lead the model to think in a different way, which is what STORYCODER achieves.

