# OpenReview forum: "StoryCoder: Bridging Narratives and Formality for Code Generation in LLMs"
_ICLR.cc/2026/Conference — ICLR 2026 Conference Withdrawn Submission_

### Official Review · Reviewer_LBqA · 2025-10-24

**Soundness:** 2
**Presentation:** 2
**Contribution:** 2
**Rating:** 4
**Confidence:** 3

**Summary:**

This paper introduces STORYCODER, a narrative-based prompting framework for code generation with LLMs. The key idea is to reformulate short, instruction-like programming problems into natural-language narratives, aiming to enhance contextual understanding and algorithmic reasoning. The method converts each problem into a structured story composed of three sections: Task Overview, Constraints, and Example Input/Output. Each reformulation is paired with a narrative genre (e.g., fantasy adventure) and an algorithmic category (e.g., dynamic programming). The authors claim that these narratives help LLMs “integrate” fragmented information and reason more effectively, leading to better code solutions.

**Strengths:**

1. Clarity: The paper is generally well-written, structured clearly, and easy to follow. Figures (especially Fig. 1 and Fig. 2) provide an intuitive understanding of the proposed pipeline.
2. Empirical thoroughness: The experiments cover multiple benchmarks and a broad range of models. The authors also conduct detailed analyses of algorithm coverage, coherence ablations, and genre misalignment, which show effort toward understanding why the method works.

**Weaknesses:**

1. Naive methodological design and Limited causal attribution: The central contribution—reformulating code problems into narratives—is conceptually shallow. While improvements are reported, it remains unclear why narrative reformulation helps.
Gains might simply result from longer prompts or additional redundancy, rather than from true “integrative reasoning.”
2. Overstated cognitive connection: The discussion invokes cognitive science theories (e.g., mental models, analogical reasoning) without providing empirical evidence that such cognitive mechanisms are truly mirrored in LLM behavior. The analogy to human reasoning feels more rhetorical than scientific.
3. Evaluation scope and baselines: Baselines such as paraphrasing, chain-of-thought (CoT), or planning-style prompting are mentioned but not fully compared under equivalent prompt lengths or instruction structures. Moreover, the authors use repeated sampling as the main baseline, which is weak compared to recent structured reasoning or code planning methods.

**Questions:**

1. Prompt length control: Have the authors controlled for the increased token length between narrative and baseline prompts? Could the improvement stem simply from more tokens and richer context rather than from narrative structure itself?

2. Baseline diversity: Why are more competitive prompting methods (e.g., program sketch prompting, self-planning, step-by-step reasoning) excluded? Would STORYCODER still outperform these under equivalent length and structure?

---

> ### Author Response · Authors · 2025-11-24
>
> We thank the reviewer for the thoughtful feedback. We appreciate the recognition of the paper’s clarity, the helpfulness of the figures, and the thoroughness of our experiments and analyses. Below are our responses to the comments.
>
> **W1. Naive design and limited causal attribution**
>
> **1. Causal mechanism of StoryCoder**
>
> We appreciate the reviewer’s concern regarding the causal mechanism behind the improvements. In the revision, we significantly strengthen this part of the analysis.
>
> First, **we have included the full narrative-solution examples in Figure B.2 and provide annotated explanations** indicating which specific narrative elements contributed to reducing implementation errors or guiding the solver toward the correct answer.
>
> In addition, we propose the following intuitions for why StoryCoder improves performance:
>
> (i) **Narratives align better with LLMs' pretraining distribution**, which is dominated by descriptive and story-like text. This provides the model with a more familiar linguistic structure and enables more coherent reasoning.
>
> (ii) **Narratives reorganize the problem into a clearer**, more solvable structure by turning scattered constraints and abstract rules into a grounded, interpretable description that helps the model identify the appropriate algorithmic pattern.
>
> (iii) **Narratives induce a more linear and model-friendly reasoning flow** by outlining a natural step-by-step progression and reducing the model's tendency to take incorrect shortcuts during implementation.
>
> We have incorporated these intuitions into the revision (Appendix B.3.)
>
> **2. Length-controlled comparison**
>
> Second, to separate the effect of narrative structure from simple prompt length, we will add length-controlled baselines:
>
> (1) concatenated rephrase variants without narrative structure,
>
> (2) random filler expansions of matched length.
>
> These comparisons will directly show that improvements do not arise from increased length or redundancy alone.
>
> We will complement these results with the existing analyses (algorithm-choice distributions, error reduction, and back-translation consistency), which collectively demonstrate that the gains reflect a genuine shift in reasoning rather than superficial prompt expansion.
>
> These revisions will substantially clarify why narrative reformulation helps, addressing the reviewer’s concern about causal attribution.
>
> **W2. Overstated cognitive-science connections**
>
>
> We acknowledge the reviewer’s concern that the cognitive science framing may appear overstated. Our intention was not to claim that LLMs exhibit human-like cognitive mechanisms, but rather to provide an intuitive explanation to help readers understand the motivation behind narrative reframing. In the revision, we will clarify that these references are purely conceptual analogies and not empirical claims about LLM cognition, and we will reduce the emphasis on cognitive science theory to avoid unintended implications.
>
> **W3. Insufficient evaluation scope and weak baselines**
>
> We acknowledge the reviewer’s concern about the breadth of baseline comparisons. The results for paraphrasing and Chain-of-Thought baselines are already included in Appendix Table A.2, and we will revise the main text to reference them more clearly. To isolate the effect of narrative structure from prompt length, we will additionally include length-controlled baselines as described in our response to Weakness 1.
>
> Furthermore, we will extend our evaluation by adding recent structured reasoning and planning-based baselines, including Structured CoT (SCoT) and Self-Planning Code Generation, under matched prompt lengths and settings.
>
> **Q1. Prompt-length controlled experiments**
>
> We will add length controlled baselines to directly address the reviewer’s concern. Specifically, we will concatenate multiple rephrase variants without introducing any narrative structure, matching the token length of StoryCoder prompts. This allows us to isolate the effect of length from the effect of narrative framing.
>
> As discussed in our response to Weakness 1, these comparisons will clarify that the improvements do not arise merely from increased prompt length or redundant context.
>
> **Q2. Missing competitive prompting baselines**
>
> We will run additional experiments with **Structured CoT (SCoT)** and **Self-Planning Code Generation** under matched conditions, and include the results in the revised manuscript.

---

> > ### Comment · Reviewer_LBqA · 2025-11-24
> >
> > I appreciate the authors' detailed explanations. The clarification of the causal mechanism makes sense to me. No more questions on this topic.
> >
> > The authors mentioned that they would conduct additional experiments (a length-controlled comparison, a comparison between SCoT and Self-Planning Code Generation). I would like the authors to provide brief results of these experiments during the rebuttal period; after that, I am willing to raise my score.

---

> > > ### Author Response · Authors · 2025-11-25
> > >
> > > Thank you for the encouraging response. We appreciate your acknowledgement of our clarifications. We are currently running the additional experiments (length-controlled comparison, SCoT, and Self-Planning Code Generation), and we will provide the corresponding results during the rebuttal period.

---

> ### Author Response · Authors · 2025-12-03
>
> We address the additional experiments and questions as follows:
>
> **1. Additional baseline comparisons**
>
> We have completed additional comparisons requested by the reviewer. Specifically, we evaluated CoT, Structured CoT (SCoT), Story-of-Thought (SoT), and paraphrasing-based baselines, including the length-controlled concatenation setting, under matched conditions. We have incorporated the results into the revised manuscript (Appendix A.3, Tables A.4 and A.5). These experiments further demonstrate that **simple increases in prompt length do not lead to performance gains**, and consistently show that StoryCoder outperforms all comparison methods.
>
> **2. Expanded related work**
>
> We have also expanded the related work section to include broader prompt-optimization literature and clarified how these approaches differ from StoryCoder’s narrative-based reformulation.

---

### Official Review · Reviewer_nJ67 · 2025-10-27

**Soundness:** 3
**Presentation:** 2
**Contribution:** 2
**Rating:** 4
**Confidence:** 4

**Summary:**

The paper introduces StoryCoder, a framework to reformulate code generation prompts into richer, contextual expression. The experiments show that this translation leads to performance improvements across various benchmarks.

**Strengths:**

- StoryCoder is a simple prompting method to translate coding instructions into narrative-style instructions. The paper is clear written and presents a wide range of experiments to support its effectiveness.
- The results show significant performance improvements over using classical coding instructions, especially in settings where a proprietary LLM is used for translating into narrative-style problems and a smaller language model is employed as the solver model.
- Translating code instructions to narrative-style prompts effectively helps the language model to choose the appropriate algorithm to solve the underlying problem. Further, it additionally reduces the error rates when it comes to implementing the corresponding problem.

**Weaknesses:**

- The paper may be considered incremental.
  - This problem of translating coding challenges into narrative-style text has been already studied in Haller et al. (2024) (https://aclanthology.org/2024.lrec-main.1111.pdf). The findings from this paper are not discussed and should be at least mentioned in the related work section for self-containment.
  - I find it confusing that the authors explain in Section 4.2. which model actually writes the narratives because I was assuming until here it is also the model that solves the coding task. I would rather read this in the methodology section that a proprietary model required and that discuss the downsides what happens if the narrative-translation model is smaller. Given that, the method heavily relies on a strong LLM in the first place. When looking at the results, the only benefit is then for cases where we want to use smaller models, however, we still need a strong LLM to translate into narrative-style problems.

- The evaluation / experiments may be considered to have flaws:
  - The authors filter random subsets of the original benchmarks and I don't see a reason why to do so. Further, there is no explanation what kind of question, e.g. in HumanEval are considered invalid.
  - The results in Table 1 depict scores for a combined settings, namely, using both, narrative-style and original instructions. While the authors account for this in the baseline approach by sampling twice as much results, I would like to see how the proposed approach - transforming original instructions into narrative-style instructions - performs on its own.
  - The results are good for settings in which we use a strong LLM anyway and then want to use smaller models as solvers. The results of StoryCoder are not improving compared to when using a strong LLM directly (Gemini 2.5 Flash, GPT-4.1).

- Selected conclusions derived by authors may be considered incorrect or incomplete:
  - The coverage ratio in Figure 4 is defined as "the fraction of problems with at least one correct solution", so basically the pass@k metric. I would rename this for better understanding.
  - I think the key statement in line 377 is overestimating some takeaways, e.g., I don't think the authors show until here that the generated code is actually more structured and follows a stepwise design. Further, an increase in correct choice of algorithms logically means a reduction in implementation error rate. However, I find it here confusing if we are actually looking at the ratio of chosing the correct algorithm (according to the text) or at the ratio of solving the actual problem (according to the Figure). If it is the ladder, the solve ratios are way higher than in Table 1, which is confusing.
  - Section 5.4 does not mention a complete list of categories to choose from and would help the reader to understand why certain categories have been selected or chosen by the authors.

**Questions:**

- Clarification on lines 159-165: How does it differ? I assume you refer to repeatedly sampling from Section 3.1. If so, I suggest to rephrase this part because using repeatedly sampling for both, instructions and answers, may be confusing.
- Fix Mistral AI citation in line 206/207
- The authors often refer to additional information in the Appendix, however, the Appendix is not present.

---

> ### Author Response · Authors · 2025-11-24
>
> We thank the reviewer for the thoughtful feedback. We appreciate the positive assessment of our narrative-based prompting approach, the clarity and breadth of our experiments, and the observed gains in algorithm selection and implementation accuracy. Below are our responses to the comments.
>
> **W1. Missing discussion of prior narrative-style code generation work**
>
> We thank the reviewer for pointing out the relevance of PECC (Haller et al., 2024). In the revision, we will incorporate this work into the related work section and clarify the distinction between PECC and our approach.
>
> While **Haller et al.** focus on evaluating LLMs’ ability to interpret prose-style, narrative-embedded coding challenges and extract problem requirements, our work differs in both goal and methodology: **StoryCoder** investigates how transforming neutral coding tasks into narratives influences algorithmic reasoning, coherence, and error reduction. Furthermore, unlike PECC, which evaluates narrative problems as given, our setting explicitly analyzes how different narrative transformations (variants, genres, coherence manipulations) affect downstream code generation. These conceptual and experimental differences will be clarified in the revision.
>
> **W2. Unclear explanation on model choice and dependence on strong LLMs**
>
> We acknowledge the reviewer’s concern regarding which model serves as the narrative generator. In the revision, we have separated the two settings in the methodology section (Section 3.2):
>
> (1) self-solving (f_narr = f_solve) and
>
> (2) cross-model (f_narr ≠ f_solve).
>
> We also agree that using a smaller model for narrative generation can limit the benefits of the method. In line with this, we explicitly discussed in the limitation section that open-source narrative generators show reduced gains on harder benchmarks. This is expected, as stronger LLMs naturally produce more coherent and information-preserving narrative reformulations, which in turn provide more reliable guidance for code generation. To make this distinction clearer, we will additionally provide paired examples of narratives written by open-source vs. closed-source models in the revised appendix.
>
> Importantly, **StoryCoder does not rely on strong LLMs to be effective.** Table 1 shows that even the closed-source self-solving models (f_narr = f_solve) achieve measurable gains on both LiveCodeBench and CodeForces. Moreover, Table 3 demonstrates that when both f_narr and f_solve are small open-source models, StoryCoder still improves performance on all benchmarks. This confirms that the method remains beneficial even when no strong LLM is used.
>
> **W3. Unjustified benchmark filtering criteria**
>
> We have revised the data filtering criteria originally described in Appendix B.2.
>
> For **HumanEval**, we filtered out problems where no input/output examples could be reliably extracted from the question text (e.g., samples lacking a function name-based usage example). Such problems cannot be evaluated under the execution-based framework used across our benchmarks, and therefore are treated as invalid.
>
> For **LiveCodeBench**, we did not randomly filter the dataset; instead, we used the latest release (v6, Jan–Apr 2025, 175 problems) to minimize pretraining contamination. Because LiveCodeBench is released chronologically, this subset contains the most recent problems and is least likely to appear in the training data of Gemini, GPT-4.1-mini, or Claude-3.5-Haiku, ensuring a fair comparison across all models.
>
> For **CodeForces**, executing all 10k problems is computationally expensive. We therefore filtered by (i) rating-based difficulty, (ii) text length, and (iii) availability of input/output formatting and generated_checkers. The resulting 265 problems are comparable in scale to LiveCodeBench (175) and HumanEval (105), enabling balanced evaluation cost without favoring any model.
>
> To address concerns about long-problem bias, we have additionally evaluated StoryCoder on a dedicated subset of long CodeForces problems; the results have been added in the revision (Appendix B.2 and Table B.1).
>
> **W4. Lack of standalone narrative-only performance results**
>
> Table 1 combines the narrative-only and narrative+original question settings because StoryCoder treats the original question as optional auxiliary context, as illustrated in Figure 1. Both are valid forms of the same narrative transformation pipeline, so the pass@10 metric reflects performance over this unified input space.
>
> Still, we agree that showing the two variants together may make interpretation less direct. In the revision, we have provided separate tables reporting pass@5 for the narrative-only and narrative+question settings in Appendix A.2 and Table A.3. Both variants consistently outperform repeated sampling, and presenting them individually will make the effectiveness of the narrative-only transformation clear.

---

> ### Author Response · Authors · 2025-11-24
>
> **W5. No gains over strong LLMs in self-solving settings**
>
> Thank you for raising the concern about whether a strong model is required to improve the performance. After re-running the experiments, we found that even when a small open-source model serves as both the narrative generator and the solver, performance improves across all three benchmarks, including LiveCodeBench. This updated result shows that the earlier concern about reduced gains with small narrative generators does not hold. Strong models still produce higher-quality narratives and yield larger improvements, but StoryCoder remains consistently beneficial even in fully small-model self-solving settings, without relying on any proprietary LLM.
>
> We have updated the manuscript (Section 4.2) to reflect these findings and highlighted that StoryCoder does not require a strong narrative generator to be effective. The performance gains come from narrative restructuring rather than from stronger models' additional knowledge alone.
>
> We also clarify that the goal of StoryCoder is not to transfer the problem-solving ability of strong models into smaller ones. Instead, the method aims to enhance a model’s own reasoning by reframing the input in a narrative structure. **For this reason, the appropriate comparison is not between (strong solver + repeated sampling) and (small solver + StoryCoder), but rather between repeated sampling and StoryCoder using the same solver.** Across all settings, including the updated small model self-solving experiments, StoryCoder consistently outperforms the baseline. These results further demonstrate that the improvements arise from structural reframing rather than dependence on a strong narrative generator.
>
> **W6. Ambiguous naming of coverage ratio (pass@k equivalence)**
>
> We agree that the term "coverage ratio" may be confusing, as it is equivalent to the pass@k definition (the fraction of problems with at least one correct solution). We have added this explanation to Section 4.1 in the revision.
>
> **W7. Overstated claims and inconsistent metric interpretation**
>
> Thank you for the detailed feedback. We agree that the statement in line 377 may be overstated given the evidence shown so far. Figure 3 alone does not fully demonstrate stepwise or more structured code generation. In the revision, **we tempered the claim and combined the Figure 3 analysis with full narrative-solution examples (Figure B.2) to clearly illustrate how narrative framing supports more structured reasoning.**
>
> Regarding the reviewer’s confusion, Figure 3(i) indeed reflects the ratio of successfully solving the problem, not the ratio of choosing the correct algorithm. We have corrected the figure caption and text accordingly.
>
> The difference between the solve ratios in Figure 3 and the scores in Table 1 comes from two factors. First, Table 1 reports pass@10, while Figure 3 computes the average correctness over individual responses. Second, as noted in lines 403-406 in revision, Figure 3 excludes trivial cases in which all 10 responses (original or narrative) are either entirely correct or entirely incorrect, since such cases do not inform the comparative analysis. Because the scoring definitions and the evaluated subsets differ, the absolute numbers in Table 1 and Figure 3 are not directly comparable.
>
> We will clarify these distinctions in the revision.
>
> **W8. Incomplete list of categories for Section 5.4 analysis**
>
> We will clarify the construction of genre categories in Section 5.4. The full list of genre candidates used in our analysis is already included in Appendix Table B.1, but we agree that the selection process should be made more explicit. We also acknowledge the concern regarding the reproducibility of misaligned genres. In the revision, we will document the procedure clearly:
>
> (1) starting from the genre names produced by the models (Figure 6(a)), we prompt GPT to generate dissimilar genre candidates, and
>
> (2) manually refine them to form the misaligned genre set $\mathcal{G}_{\text{mis}}$.
>
> This process will be described in Appendix B.4.
>
> These additions will make the construction of both aligned and misaligned genre categories transparent and reproducible.
>
> **Q1. Clarification of repeated sampling distinction (lines 159–165)**
>
> To avoid confusion, we have revised the phrasing in line 162 by replacing "repeated sampling" with “narrative variant generation,” ensuring a clear distinction from the repeated sampling baseline described in Section 3.1.
>
> **Q2 & Q3. Incorrect Mistral AI citation & missing appendix**
>
> We have revised the reference accordingly. We would appreciate it if you could kindly check the appendix file available in the Supplementary Material section on the OpenReview page. **Currently, in the revised PDF, the main paper and the appendix are included in a single file.**

---

> > ### Comment · Reviewer_nJ67 · 2025-11-27
> > **Reviewer Response**
> >
> > Thank you very much for the clarifications. I read the updated paper again, and I consider my presentation concern as resolved and will update the scores accordingly.
> >
> > However, I am still not completely convinced about the main contribution. I acknowledge the broad evaluation across many models but reporting scores mostly at pass@10 seems more favorable, while the baseline performs better for pass@[1–4] and this also appears to be more efficient in practice.
> >
> > Also, I believe several core conceptual questions remain unresolved. For example, the current framing suggests a generalization claim that coding problems should always be translated into narrative-based forms, which may be considered an incremental contribution. While the explicit definition of genres such as science fiction may be interesting as an independent contribution, the current experimental analysis (e.g., PCA highlighting misaligned topics or choosing layouts rather than genres (such as hospital forms)) does not sufficiently demonstrate why genre-specific reformulations are necessary.
> >
> > Further, in the current revision, it is unclear how paraphrasing differs from StoryCoder, since StoryCoder is actually a paraphrase-like transformation of the original coding problem. This makes it difficult to interpret why the paraphrasing baseline performs so poorly and raises concerns that the comparison may not be fair or sufficiently disentangled.
> > Finally, the dependence on a predefined set of algorithm categories remains unclear. Is this taxonomy fixed? How does the method handle problems that fall outside these categories, or overlap multiple categories?

---

> ### Author Response · Authors · 2025-12-01
>
> We appreciate the reviewer’s thoughtful reassessment and score update, and we thank you for the insightful comments and the opportunity to clarify our contribution. We address the reviewer’s comments as follows:
>
> **1. Limited Improvements at small k (Pass@1–4)**
>
> Each narrative has a coherent point of view and story structure. Therefore, with only a few samples (pass@1-4), the benefits of the narrative reformulation could be diluted because the variety of generated narratives is limited. However, as more narrative variants are produced, the chances increase that one of them will offer a useful reasoning path for solving the problem. A similar observation has been discussed in prior work [1].
>
> As a result, the advantage may be difficult to observe at small k, but becomes more evident as k grows, reflecting how StoryCoder expands perspectives and narrative expressions to widen the model’s search space for identifying a correct solution. This behavior can be clearly observed on challenging benchmarks such as LiveCodeBench and CodeForces, as shown in the pass@k curves in Figure A.1 and A.2.
>
> [1] Planning In Natural Language Improves LLM Search For Code Generation, ICLR 2025 (https://arxiv.org/abs/2409.03733)
>
> **2. Generalization claim about narrative-only reformulation**
>
> We do not claim that all coding problems should be translated into narrative form. StoryCoder is a framework designed to guide LLM reasoning in code generation tasks that require structured inference, and **proposing a universal transformation rule is not the goal of this work.**
>
> **3. Concerns about the necessity and validity of genre- and algorithm-based reformulations**
>
> The algorithm and genre tags are not core determinants of method performance. They were introduced as auxiliary elements to identify the sources of performance gains and to systematically examine how the model’s linguistic expressions vary. Both the genre and algorithm tags function only within the narrative generator as high-level organizational hints and are never exposed to the solver itself. The **algorithm taxonomy is a fixed set**, as clarified in the revision (Section 3.2), and simply provides broad categories that cover common solution patterns in coding benchmarks. Consequently, performance is robust to the precision or single-category assignment of these labels.
>
> To address the reviewer’s concern regarding the necessity of algorithm- or genre-specific reformulations, we have additionally evaluated an **algorithm- and genre-free version** of StoryCoder that applies only the structural transformation guidelines without any explicit algorithm or genre labels (Appendix A.2 and Table A.4). This variant also consistently improved performance over the original baseline, and in some cases even outperformed the full narrative version.
>
> These results demonstrate that the primary mechanism behind the improvement lies in strengthening the model’s reasoning trajectory through structural reorganization of the context, rather than any genre specification or reliance on fixed algorithm categories.
>
> **4. Distinction from simple paraphrasing**
>
> We have added a paraphrase example in the revision (Appendix A.2 and Figure B.3). As illustrated, paraphrasing changes only the surface wording and preserves the original structure, so it does not lead the model to think in a different way, which is what StoryCoder achieves.

---

> ### Author Response · Authors · 2025-12-03
>
> We have incorporated PECC (Haller et al., 2024) into the revised related work section and **clarified how its objectives and evaluation setup differ from ours.** The revision now clearly distinguishes PECC’s focus on interpreting narrative-guided, simpler coding benchmarks from StoryCoder’s goal of analyzing how narrative transformations influence algorithmic reasoning and downstream code generation performance.

---

### Official Review · Reviewer_dpFb · 2025-10-28

**Soundness:** 2
**Presentation:** 1
**Contribution:** 2
**Rating:** 2
**Confidence:** 3

**Summary:**

The paper introduces StoryCoder, a narrative-based prompting framework that reformulates coding problems into structured natural-language stories to improve LLM code generation. StoryCoder constructs each narrative with three components: (1) Task Overview, (2) Constraints, and (3) Example Input/Output. The narrative is then used by a solver model to generate code. Experiments conducted on HumanEval, LiveCodeBench, and CodeForces demonstrate that narrative prompting consistently enhances model performance, leading to higher pass@10 accuracy across all evaluated models. The authors further analyze 1) algorithm selection fidelity, 2) implementation error reduction, 3) narrative coherence effects, and 4) genre alignment influence on model performance.

**Strengths:**

1. Reframing code generation as storytelling is an interesting idea that bridges narrative cognition and programming reasoning, grounded in cognitive theories like mental models and analogical mapping.
2. The evaluation covers 11 LLMs, including both closed-weight and open-weight models, tested across three benchmarks of varying difficulty to ensure strong validity.
3. The paper conducts in-depth analyses beyond task performance, decomposing results into factors like algorithm selection, implementation errors, and correctness. It further examines coherence and genre effects through controlled narrative variants and uses back-translation analysis to show genuine shifts in model reasoning, not just output variation.

**Weaknesses:**

**Major Concerns:**
1. In the introduction, the authors introduce the terms “algorithm” and “genre” in the context of narrative generation. However, these concepts are not formally defined in the methodology section. Later, the paper analyzes the effectiveness of each component, which becomes difficult to follow without clear definitions of these terms.
2. The descriptions of Repeated Sampling (RS) in Sections 3.1 (line 140) and 3.2 (line 162) are inconsistent. In Section 3.1, the authors define RS as sampling multiple responses for the same input. In Section 3.2 (line 162), they define RS again as generating multiple narratives that represent different input variants.
3. Following the previous confusion, Section 4.1 (line 247) states that the authors sample twice as many outputs, which adds more ambiguity. The reference to Equation 4 in this context is also missing.
4. The table 1 shows the results of open-source LLMs of combined setting for both narrative-only and narrative-question pair inputs, this introduce additional confusion of why doing this? Making the results more harder to interpret.
5. In table 1, the author only shows results of repeated sampling and their proposed narrative prompting, why there is no results of paraphrasing and Chain-of-Thought (CoT) which are all introduced in Section 3.1?
6. For the baselines, the author only includes non–prompt-optimization methods. Why were other prompt reformulation baselines, such as Structured Chain-of-Thought (SCoT, https://arxiv.org/abs/2305.06599), not included? SCoT was already mentioned by the author in the related work section. And I believe there are many other valid prompt-optimization for code generation baseline could be included such as Self-Planning Code Generation with LLM (https://dl.acm.org/doi/pdf/10.1145/3672456)
7. In the evaluation, the authors used a setting where narratives were generated by one model (Gemini-2.5-Flash), and each open-source model generated code based on the same narratives, as shown in Table 1. Later, they again argued that the self-solving setting is not suitable for open-source models because these models cannot produce valid narratives, and thus they used a different model (Qwen-2.5-32B) to generate the narratives. However, it is unclear why this additional setting is needed, since the same condition—using shared narratives—is already included in the table 1. This makes the logic difficult to follow.
8. In Table 2, the author reports the proportion of valid narratives generated by each open-source model but does not clearly define what constitutes a valid narrative or explain how narrative validity was assessed. In addition, it is unclear what the validity rate of narratives is for the self-solving LLM shown in Table 1.
9. In Section 5, the lack of formal definition of algorithm and genre in narrative generation makes the analysis hard to follow. For example, “Misaligned genres” are manually chosen by the authors. There’s no objective way to determine what counts as aligned vs. misaligned, which hurts reproducibility.
10. In Section 5, most of the analysis were based on LLM-as-a-Judge, there is a lack of human validation to show the correctness of LLM-as-a-Judge for each analysis.

**Minor Concerns:**
1. The date filtering criteria are missing from the main content, which I think is important for readers to understand the validity of the results. (The appendix is also missing from the submission.)
2. I believe there are many other related works on prompt optimization beyond code generation, which could be useful for better understanding this research direction and strengthening the motivation.

**Questions:**

1. What is the search space of algorithm and genre in narrative generation?
2. For baselines, why do you not include some reasoning model for comparison? I believe reasoning model may already has the capability to reflect on the input question and reformulate it in their internal reasoning steps.
3. Could you explain why including a story improves the quality of output code? From the example in Figure 2, the generated story does not seem to contain information that directly helps solve the coding problem (just my opinion). It might be clearer if the authors highlight which parts of the story provide relevant information.

---

> ### Author Response · Authors · 2025-11-24
>
> We thank the reviewer for the thoughtful feedback. We appreciate the recognition of our storytelling-based framing, the broad evaluation across 11 models and multiple benchmarks, and the depth of our analyses on algorithm choice and narrative effects. Below are our responses to the comments.
>
> **W1. Undefined algorithm and genre concepts**
>
> Thank you for pointing out the ambiguity regarding the terms "algorithm" and "genre" in our narrative generation framework. In the revision, we have explicitly defined both concepts in Section 3.2:
>
> - **Algorithm** is defined as the high-level problem-solving strategy underlying the coding task that the narrative is instructed to reflect, chosen from the eight predefined categories in Appendix Figure B.1.
>
> - **Genre** is defined as the narrative style, which is not chosen from a fixed list but left for the model to select freely in a way that aligns with the problem and the chosen algorithm.
>
>
> **W2. Inconsistent definitions of Repeated Sampling**
>
>
> Previously, in Section 3.2, the term "repeated sampling (RS)" was mistakenly reused to describe generating narrative variants. To avoid this ambiguity, we have revised this and all related expressions to use the term "narrative variant generation", ensuring that RS is used exclusively to denote sampling multiple solver outputs for a fixed input.
>
> **W3. Ambiguous output sampling explanation**
>
> In our experiments (Table 1, 3, A.1, and A.2), we set $N$ (the number of narrative variants) = 5. For pass@10 computation, StoryCoder generates
>
> - five narrative variants $\{\mathcal{N}_i^j\}_{j=1}^5$, and
>
> - five narrative variants + original question \{\mathcal{N}_i^j, Q_i\}_{j=1}^5, yielding 10 total results per question $Q_i$.
>
> In the RS setting, we perform 10 repeated samples to match this count. We have clarifyied this experimental setup in the revision (Section 4.1.)
>
> **W4. Confusing combined narrative-only and narrative+question results**
>
>
> Table 1 combines the narrative-only and the narrative+original question settings because StoryCoder treats the original question as optional auxiliary context, as illustrated in Figure 1. Both forms are valid realizations of the same narrative transformation pipeline, and the pass@10 metric reflects performance over this unified input space.
>
> Nevertheless, we acknowledge that aggregating the two input modes may make interpretation less direct. To address this, we have added separate tables reporting **pass@5 for the narrative-only and narrative+question settings individually** (Appendix A.2 and Table A.3). Both settings consistently outperform the baseline, and these additional results will make the comparison more transparent.
>
> **W5. Missing paraphrasing and CoT baselines in Table 1**
>
> The results for paraphrasing and Chain-of-Thought (CoT) are already included in Appendix Table A.2. We have revised the main text to make this reference explicit and ensure that readers can easily find these comparisons in Section 4.2.
>
> **W6. Lack of prompt-optimization baselines (e.g., SCoT, self-planning)**
>
> We thank the reviewer for this suggestion to broaden our comparisons. We will additionally evaluate prompt-reformulation baselines such as SCoT and Self-Planning Code Generation under the same experimental conditions and include the results in the revised manuscript.
>
> **W7. Unclear rationale for additional narrative-provider setting**
>
> Table 1 (rows 5–13) evaluates open-source models using shared narratives generated by Gemini, enabling a controlled comparison under a high-quality narrative setting. In contrast, Table 3 serves a different purpose: it assesses whether StoryCoder remains effective when open-source models themselves generate the narratives.
>
> To analyze the open-source self-solving behavior, we additionally use Qwen 2.5 32B Instruct, Mistral-Small 24B Instruct, and Gemma 2 27B Instruct as $f_{\text{narr}}$ in Table 3, since these models achieve the highest valid narrative rates among open-source generators.
>
> The two shared-narrative experiments address distinct research questions:
> (i) **controlled comparison under high-quality narratives** (Table 1), and
> (ii) **evaluation of StoryCoder’s robustness under open-source models' self-solving setting** (Table 3).

---

> ### Author Response · Authors · 2025-11-24
>
> **W8. Undefined narrative validity criteria**
>
> We thank the reviewer for pointing out the lack of clarity regarding how narrative validity was assessed. We will add the precise criteria used in Table 2: narrative outputs were deemed invalid if (i) the sequence length was fewer than 50 tokens (near-empty content), or (ii) the sequence length exceeded 99% of the model’s maximum generation limit, which we confirmed corresponds to degenerate looping or meaningless token repetition. These filtering rules have been included in the revision (Appendix B.4).
>
> Regarding the reviewer’s question about the self-solving LLMs in Table 1, we clarify that the **three closed-source models (Gemini-2.5-Flash, GPT-4.1-mini, and Claude-3.5-Haiku) produced 100% valid narratives.** Because validity was perfect in this setting, we omitted the number in the main text. For clarity, we have reported these results in the revision (Appendix B.4).
>
> **W9. Unclear algorithm/genre definitions and misaligned genres criteria**
>
> As noted in Weakness 1, we will provide formal definitions of algorithm and genre in the methodology section.
>
> We also acknowledge the concern regarding the reproducibility of misaligned genres. In the revision, we will clearly document the construction procedure:
> (1) starting from the genre names actually produced by the models (Figure 6(a)), we prompt GPT to generate dissimilar genre candidates, and
> (2) manually refine them to form the misaligned genre set $\mathcal{G}_{\text{mis}}$.
> This process will be described in Appendix B.4.
>
> To further provide an objective distinction between aligned and misaligned genres, we will report inter- and intra-cluster distances in the genre embedding space (Figure 6(b), PCA projection), showing that aligned genres form a cohesive cluster while misaligned genres occupy a separate region.
>
> **W10. Lack of human validation for LLM-as-a-Judge analyses**
>
> We will address this concern by adding a human evaluation on a representative 5–10% subset of all benchmarks. Human annotators will verify the judgments used in Section 5, and we will report agreement statistics to confirm the reliability of the LLM-as-a-Judge analyses.
>
> **MW1. Missing data filtering criteria in main text**
>
> Thank you for raising the concern about validity. We have revised the data filtering criteria described in Appendix B.2 with a more detailed explanation.
>
> We used only the latest **LiveCodeBench** subset (v6, Jan–Apr 2025), consisting of 175 samples, to minimize the risk of pretraining contamination for all evaluated models. Since LiveCodeBench is released chronologically, this subset consists of the most recently published samples. Considering the knowledge cutoffs of Gemini-2.5-Flash (Jan 2025), GPT-4.1-mini (Jun 2024), and Claude-3.5-Haiku (Jul 2024), release-v6 is the split with the lowest likelihood of appearing in the training data of any of these models. Therefore, using this subset ensures a fair comparison across models.
>
> For **CodeForces**, evaluating all 10k samples with execution-based scoring is expensive. We therefore filtered the dataset by (i) rating-based difficulty, (ii) text length, and (iii) input/output formatting availability. This produced a set of 265 problems, slightly larger than LiveCodeBench (175) and HumanEval (105), allowing us to balance evaluation cost across the three benchmarks. This design choice is purely computational and does not favor any specific model or method.
>
> Furthermore, to address the reviewer's concern regarding potential bias on long problems, we have evaluated StoryCoder on a subset of CodeForces problems with longer lengths (CodeForces-L). Our additional experiments confirm that StoryCoder consistently improves performance even on the longer subset, CodeForces-L, and these results have been included in the revision in Appendix B.2 and Table B.1.
>
> **MW2. Incomplete coverage of broader prompt-optimization literature**
>
> We agree that broader prompt-optimization literature is relevant to this work. We will expand the related work section to include additional prompt optimization methods and clarify how these approaches differ from StoryCoder’s focus on narrative reformulation.
>
> **Q1. Search space of algorithms and genres**
>
> Algorithm selection is restricted to the eight predefined categories listed in Appendix Figure B.1, from which the model chooses the category most appropriate for the coding task.
>
> For genre, we do not specify a predefined list; the model freely selects a narrative style. In the revision, we explicitly defined the genre search space as the set of naturally emerging genre clusters observed in the embedding space (Section 5.4).

---

> ### Author Response · Authors · 2025-11-24
>
> **Q2. Absence of reasoning-model baselines**
>
> We acknowledge the reviewer’s point regarding reasoning models. While full evaluations were infeasible under our computational budget, we will include additional subset experiments using strong reasoning LLMs in the revision. We will report the results after finalizing our experiments and this will allow us to empirically compare narrative prompting with internal reasoning and clarify the distinctions between the two approaches.
>
> **Q3. Mechanism by which stories improve code quality**
>
> We agree that Figure 2 alone may not clearly convey why the narrative improves code quality, since the narrative content is partially omitted for space. In the revision, we have provided **full example of narrative and model solution pairs in Appendix B.3 and Figure B.2**, and explicitly highlighted which narrative elements contributed to guiding the model toward the correct answer.
>
> In addition, we propose the following intuitions for why StoryCoder improves performance:
>
> (i) **Narratives align better with LLMs' pretraining distribution**, which is dominated by descriptive and story-like text. This provides the model with a more familiar linguistic structure and enables more coherent reasoning.
>
> (ii) **Narratives reorganize the problem into a clearer**, more solvable structure by turning scattered constraints and abstract rules into a grounded, interpretable description that helps the model identify the appropriate algorithmic pattern.
>
> (iii) **Narratives induce a more linear and model-friendly reasoning flow** by outlining a natural step-by-step progression and reducing the model's tendency to take incorrect shortcuts during implementation.
>
> We have incorporated these intuitions into the revision in Appendix B.3.
>
> These revisions will make the mechanism more interpretable and demonstrate that the benefit does not rely on additional problem information, but on restructuring the original constraints into a more coherent reasoning process.

---

> ### Author Response · Authors · 2025-12-03
>
> We address the additional experiments and questions as follows:
>
> **1. Additional Baseline Comparisons**
>
> We have completed the additional comparisons by evaluating Structured CoT (SCoT), Story-of-Thought (SoT), and length-controlled experiments and have incorporated the results into the revision in Appendix A.3, Table A.4, and A.5. These experiments consistently show that **StoryCoder outperforms the comparison methods.**
>
> **2. Expanded Related Work**
>
> We have expanded the related work section to include broader prompt-optimization literature and clarified how these approaches differ from StoryCoder's narrative reformulation.

---

### Official Review · Reviewer_AvFE · 2025-10-31

**Soundness:** 3
**Presentation:** 3
**Contribution:** 3
**Rating:** 2
**Confidence:** 3

**Summary:**

This work propose STORYCODER, a framework that transforms code generation questions into narratives (stories) before prompting LLMs. The experiments showed that prompting with natural language narratives yield substantial improvements across many LLMs on several benchmarks.

**Strengths:**

- The idea is converting questions into narratives for code generation is simple, yet quite effective.
- The improvements are substantial and consistent across many models.

**Weaknesses:**

- Although the idea of transforming questions into narratives is new for software domains, it has been explored previously in Story of thought (SoT) (Sadiri Javadi et al., 2025), which is also cited by the authors. Due to the high similarly between the two, I would expect a more thorough discussion and empirical comparison to highlight the strengths and weaknesses of each approach.
- The experimental setting is questionable. For LiveCodeBench, why did the authors only filter out 175 out of the original 1,055? Appendix B2 states that this is to avoid data contamination, but it is not convincing since StoryCoder does not perform any additional training. Similarly, CodeForces was also filtered to have 265 out of 10k samples. Given that the text length must be less than 1k, is it true that StoryCoder cannot generate reliable narratives for long problems?
- Given that the empirical evaluation is only conducted on rather small subsets of existing benchmarks, it is unclear if StoryCoder can generalize well.
- What is the running time overhead of StoryCoder compared to baselines like repeated sampling and SoT?

**Questions:**

See Weaknesses.

---

> ### Author Response · Authors · 2025-11-18
>
> We thank the reviewer for the thoughtful feedback. We appreciate the positive assessment of the simplicity and effectiveness of our approach, as well as its consistent improvements across models. Below are our responses to the comments.
>
> **W1. Insufficient comparison with related work (SoT)**
>
> We appreciate the reviewer's suggestion regarding Story of Thought (SoT). While both approaches share the high-level idea of reformulating a problem into a narrative, their goals and transformation principles differ substantially: **SoT** is designed for multiple-choice common-sense reasoning, whereas **StoryCoder** targets code-generation tasks, requiring precise preservation of constraints, I/O specifications, and algorithmic structure.
>
> Nevertheless, we agree that a more direct comparison would strengthen the contribution. We are currently conducting a comparison experiment by applying the SoT guideline to the code-generation benchmarks and reporting how its reformulation quality and downstream performance differ from StoryCoder. This will clarify the strengths and limitations of each approach in the code-generation domain.
>
> **W2. Data filtering criteria and handling of long problems**
>
> **1. Evaluation on Long Problem Descriptions**
>
> To address the reviewer's concern regarding potential bias on long problems, we have evaluated StoryCoder on a subset of CodeForces problems with longer lengths (CodeForces-L). Our additional experiments confirm that **StoryCoder consistently improves performance even on the longer subset**, CodeForces-L, and these results have been included in the revision in Appendix B.2 and Table B.1.
>
> **2. Dataset Construction and Fair Evaluation**
>
> We used only the latest **LiveCodeBench** subset (v6, Jan–Apr 2025), consisting of 175 samples, to minimize the risk of pretraining contamination for all evaluated models. Since LiveCodeBench is released chronologically, this subset consists of the most recently published samples. Considering the knowledge cutoffs of Gemini-2.5-Flash (Jan 2025), GPT-4.1-mini (Jun 2024), and Claude-3.5-Haiku (Jul 2024), release-v6 is the split with the lowest likelihood of appearing in the training data of any of these models. Therefore, using this subset ensures a fair comparison across models.
>
> For **CodeForces**, evaluating all 10k samples with execution-based scoring is expensive. We therefore filtered the dataset by (i) rating-based difficulty, (ii) text length, and (iii) input/output formatting availability. This produced a set of 265 problems, slightly larger than LiveCodeBench (175) and HumanEval (105), allowing us to balance evaluation cost across the three benchmarks. This design choice is purely computational and does not favor any specific model or method.
>
> **W3. Limited generalization due to small benchmark subsets**
>
> We understand the reviewer's concern regarding generalization. This issue is closely related to the previous comment on data filtering. As clarified above, our subsets were chosen due to contamination avoidance (LiveCodeBench) and compute constraints (CodeForces), not because StoryCoder performs poorly on the filtered-out samples. The additional longer length subset evaluation and the SoT/self-planning prompting baselines will address this concern.
>
> **W4. Unclear runtime overhead of StoryCoder compared to baselines**
>
> We thank the reviewer for raising this practical question. StoryCoder adds only a single additional step compared to the baselines: $N$ forward passes of the narrative generator to produce $N$ narrative variants, while the solver's computation remains unchanged. We will report the average narrative generation latency for both open- and closed-source models on our hardware setup, showing that this overhead is modest relative to code-generation time and does not dominate the end-to-end inference cost.

---

> ### Author Response · Authors · 2025-11-24
>
> We have completed the additional evaluation on long problem descriptions raised in Weakness 2, and we have updated both the response and the manuscript (Appendix B.2 and Table B.1) accordingly. We hope these additions address the reviewer's concern.

---

> ### Author Response · Authors · 2025-12-03
>
> We address the additional experiments and questions as follows:
>
> **1. Comparison with Story-of-Thought**
>
> We have completed this comparison experiment by applying Story-of-Thought (SoT) guidelines to our code-generation benchmarks. The results in Appendix A.3 and Table A.4 show that SoT's narrative transformations do not reliably preserve algorithmic structure or task constraints, leading to **significantly weaker performance than StoryCoder.** This confirms the conceptual differences between the two methods and clarifies the strengths of our approach in the code-generation setting.
>
> **2. Time Overhead**
>
> We have measured an average latency for narrative generation of **0.907 seconds** over 8 runs using Gemini-2.5-Flash. Moreover, because StoryCoder's narrative variant generation can be parallelized rather than executed sequentially, the additional time overhead introduced by our method is negligible.

---

### Author Response · Authors · 2025-11-18
**Author Summary Response**

We sincerely thank the reviewers for their constructive and encouraging feedback:

1. **Simplicity and Effectiveness**: StoryCoder is acknowledged as simple yet yields substantial and consistent improvements across many models.
2. **Novelty**: The narrative reframing of coding tasks is highlighted as a novel and interesting idea within code generation.
3. **Comprehensive Evaluation**: The work is recognized for its broad evaluation across 11 LLMs and three benchmarks of varying difficulty.
4. **In-Depth Analyses**: The detailed analyses of algorithm selection, error types, coherence, and genre effects are seen as helpful for understanding why the method works.
5. **Clarity and Accessibility**: The paper is described as clearly written, well-structured, and supported by intuitive figures.

We deeply appreciate these positive assessments and will provide detailed responses to each reviewer’s comment.

We will append the additional experimental results to each response as soon as the experiments are completed to support a clear discussion.

It appears that some reviewers may not have been able to access the appendix file. The appendix was submitted as a separate PDF in the Supplementary Material section, and we would appreciate it if the reviewer could refer to that file for the full details.

---

### Author Response · Authors · 2025-12-03
**Summary of Revisions for the Area Chair**

We thank the reviewers for their constructive feedback. We believe that the revisions made in response to the reviewers' feedback have substantially improved the manuscript, and we hope that it is now suitable for publication at ICLR. We summarize our overall revisions as follows:

**1. Expanded Baseline Comparisons**

We addressed the reviewer’s concern by significantly broadening the baseline set (Appendix A.3, Tables A.4–A.5).

- Added Structured CoT (SCoT) and Story-of-Thought (SoT) comparisons


- Added length-controlled paraphrase concatenation (PC) baselines


These comparisons show that StoryCoder consistently outperforms all methods.

**2. Causal Mechanism of Narrative Reformulation**

We strengthened the explanation of why narrative transformation improves model reasoning beyond surface-level rewriting (Appendix B.3).

- Added full narrative–solution examples with annotated reasoning cues

- Clarified how narratives reorganize constraints and reflect algorithmic flow

- Introduced controlled-length baselines to show that gains are not due to prompt length alone

**3. Integration of SoT and PECC**

We incorporated additional narrative-style methods into the analysis to clarify conceptual distinctions (Appendix A.3; Related Work).

- Added empirical comparison with SoT applied to code-generation tasks

- Added PECC (Haller et al., 2024) to Related Work with clarified differences in goals and evaluation setup

- Highlighted that StoryCoder targets algorithmic reasoning rather than explanation extraction

**4. Prompt-Length Control**

We tested whether longer inputs alone explain the performance improvements (Appendix A.3; Table A.5).

- Added paraphrase concatenation (PC) with matched length

- Verified that longer prompts do not achieve comparable gains

**5. Dataset Filtering and Generalization**

We clarified dataset handling choices and expanded evaluation to more challenging subsets (Appendix B.2; Table B.1).

- Explained filtering on HumanEval, LiveCodeBench, and CodeForces

- Added evaluation on long CodeForces problems

- Demonstrated robustness beyond short or simple tasks

**6. Clarification of Algorithm and Genre Tags**

We standardized and expanded the explanation of the algorithm and the genre tags (Section 3.2; Appendix A.2).

- Added formal definitions of algorithm and genre tags

- Added tag-free StoryCoder ablation

- Verified that tags are not responsible for performance gains

**7. Experimental Design Clarifications**

We improved clarity in the experimental setup and terminology (Appendix A.2; Table A.3).

- Defined repeated sampling (RS) vs. narrative-variant sampling

- Added separate evaluation for narrative-only and narrative+question settings

---

### Note · Authors · 2026-01-06

I have read and agree with the venue's withdrawal policy on behalf of myself and my co-authors.